# GRAPHDEEPONET: LEARNING TO SIMULATE TIME-DEPENDENT PARTIAL DIFFERENTIAL EQUATIONS USING GRAPH NEURAL NETWORK AND DEEP OPERATOR NETWORK

## ABSTRACT

Scientific computing using deep learning has seen significant advancements in recent years. There has been growing interest in models that learn the operator from the parameters of a partial differential equation (PDE) to the corresponding solutions. Deep Operator Network (DeepONet) and Fourier Neural operator, among other models, have been designed with structures suitable for handling functions as inputs and outputs, enabling real-time predictions as surrogate models for solution operators. There has also been significant progress in the research on surrogate models based on graph neural networks (GNNs), specifically targeting the dynamics in time-dependent PDEs. In this paper, we propose GraphDeepONet, an autoregressive model based on GNNs, to effectively adapt DeepONet, which is well-known for successful operator learning. GraphDeepONet outperforms existing GNN-based PDE solver models by accurately predicting solutions, even on irregular grids, while inheriting the advantages of DeepONet, allowing predictions on arbitrary grids. Additionally, unlike traditional DeepONet and its variants, GraphDeepONet enables time extrapolation for time-dependent PDE solutions. We also provide theoretical analysis of the universal approximation capability of GraphDeepONet in approximating continuous operators across arbitrary time intervals.

## 1 INTRODUCTION

Various physical phenomena can be expressed as systems of partial differential equations (PDEs). In recent years, there has been a growing interest in leveraging deep learning techniques to enhance the efficiency of scientific computing. The field of scientific computing plays a crucial role in approximating and simulating solutions to PDEs, making it a subject of significant importance and active research (Guo et al., 2016; Zhu et al., 2019). As the exploration of deep learning applications in scientific computing gains momentum, it opens up exciting possibilities for advancing our understanding and modeling of complex phenomena (Raissi et al., 2019; Karniadakis et al., 2021).

In recent years, operator learning frameworks have gained significant attention in the field of artificial intelligence. The primary goal of operator learning is to employ neural networks to learn the mapping from the parameters (external force, initial, and boundary condition) of a PDE to its corresponding solution operator. To accomplish this, researchers are exploring diverse models and methods, such as the deep operator network (DeepONet) (Lu et al., 2019) and Fourier neural operator (FNO) (Li et al., 2020), to effectively handle functions as inputs and outputs of neural networks. These frameworks present promising approaches to solving PDEs by directly learning the underlying operators from available data. Several studies (Lu et al., 2022; Goswami et al., 2022) have conducted comparisons between DeepONet and FNO, and with theoretical analyses (Lanthaler et al., 2022; Kovachki et al., 2021a) have been performed to understand their universality and approximation bounds.

In the field of operator learning, there is an active research focus on predicting time-evolving physical quantities. In this line of research, models are trained using data that captures the evolution of

physical quantities over time when the initial state is provided. Once the model is trained, it can be applied to real-time predictions when new initial states are given. This approach finds practical applications in various fields, such as weather forecasting (Kurth et al., 2022) and control problems (Hwang et al., 2021). These models can be interpreted as time-dependent PDEs. The DeepONet can be applied to simulate time-dependent PDEs by incorporating a time variable, denoted as $t$, as an additional input with spatial variables, denoted as $x$. However, the use of both $t$ and $x$ as inputs at once to the DeepONet can only predict solutions within a fixed time domain and they should be treated differently from a coefficient and basis perspective. FNO (Li et al., 2020; Kovachki et al., 2021b) also introduces two methods specifically designed for this purpose: FNO-2d, which utilizes an autoregressive model, and FNO-3d. However, a drawback of FNO is its reliance on a fixed uniform grid. To address this concern, recent studies have explored the modified FNO (Lingsch et al., 2023; Lin et al., 2022), such as geo-FNO (Li et al., 2022b) and F-FNO (Tran et al., 2023).

To overcome this limitation, researchers have explored the application of GNNs and message passing methods (Scarselli et al., 2008; Battaglia et al., 2018; Gilmer et al., 2017) to learn time-dependent PDE solutions. Many works (Sanchez-Gonzalez et al., 2020; Pfaff et al., 2021; Lienen & Günnemann, 2022) have proposed graph-based architectures to simulate a wide range of physical scenarios. In particular, Brandstetter et al. (2022) and Boussif et al. (2022) focused on solving the time-dependent PDE based on GNNs. Brandstetter et al. (2022) proposed a Message-Passing Neural PDE Solver (MP-PDE) that utilizes message passing to enable the learning of the solution operator for PDEs, even on irregular domains. However, a limitation of their approach is that it can only predict the solution operator on the same irregular grid used as input, which poses challenges for practical simulation applications. To address this limitation, Boussif et al. (2022) introduced the Mesh Agnostic Neural PDE solver (MAgNet), which employs a network for interpolation in the feature space. This approach allows for more versatile predictions and overcomes the constraints of using the same irregular grid for both input and solution operator prediction. We aim to employ the DeepONet model, which learns the basis of the target function's spatial domain, to directly acquire the continuous space solution operator of time-dependent PDEs without requiring additional interpolation steps. By doing so, we seek to achieve more accurate predictions at all spatial positions without relying on separate interpolation processes.

In this study, we propose GraphDeepONet, an autoregressive model based on GNNs, effectively adapting the well-established DeepONet. GraphDeepONet surpasses existing GNN-based models by accurately predicting solutions even on irregular grids while retaining the advantages of DeepONet, allowing predictions on arbitrary grids. Moreover, unlike conventional DeepONet and its variations, GraphDeepONet enables time extrapolation. Experimental results on various PDEs, including the 1D Burgers' equation and 2D shallow water equation, provide strong evidence supporting the efficacy of our proposed approach. Our main contributions can be summarized as follows:

- By effectively incorporating time information into the branch net using a GNN, GraphDeepONet enables time extrapolation prediction for PDE solutions, a task that is challenging for traditional DeepONet and its variants.

- Our method exhibits superior accuracy in predicting the solution operator at arbitrary positions of the input on irregular grids compared to other graph-based PDE solver approaches. The solution obtained through GraphDeepONet is a continuous solution in the spatial domain.

- We provide the theoretical guarantee that GraphDeepONet is universally capable of approximating continuous operators for arbitrary time intervals.

## 2 RELATED WORK

In recent years, numerous deep learning models for simulating PDEs have emerged (Sirignano & Spiliopoulos, 2018; E & Yu, 2018; Karniadakis et al., 2021). One instance involves operator learning (Guo et al., 2016; Zhu et al., 2019; Bhatnagar et al., 2019; Khoo et al., 2021), where neural networks are used to represent the relationship between the parameters of a given PDE and the corresponding solutions (Kovachki et al., 2021b). FNO and its variants (Helwig et al., 2023; Li et al., 2022a) are developed to learn the function input and function output. Building on theoretical foundations from Chen & Chen (1995), Lu et al. (2019) introduced the DeepONet model for PDE simulation. DeepONet has been utilized in various domains, such as fluid dynamics, hypersonic scenarios and

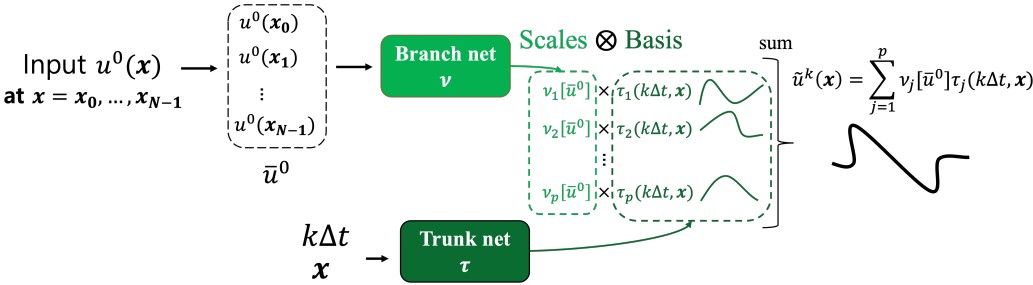

Figure 1: The original DeepONet structure (Lu et al., 2021) for simulating time-dependent PDE.

bubble dynamics predictions. Efforts to enhance the DeepONet model have led to the development of various modified versions (Wang et al., 2021; Prasthofer et al., 2022; Lee et al., 2023). Notably, Variable-Input Deep Operator Network (VIDON), proposed by Prasthofer et al. (2022), shares similarities with our approach in that it uses transformers to simulate PDEs on random irregular domains. However, it does not address the simulation of time-dependent PDEs propagating over time, which is a significant distinction in our work. Many studies also focusing on using latent states to simulate time-dependent PDEs (Mücke et al., 2021; Yin et al., 2023).

Another emerging area of research involves using GNNs for operator learning (Alet et al., 2019; Seo et al., 2019; Belbute-Peres et al., 2020; Iakovlev et al., 2020; Lienen & Günnemann, 2022; Horie & Mitsume, 2022). Li et al. (2019) introduced a graph-based neural operator model that learn a solution operators of PDEs from external forces represented as graphs. Sanchez-Gonzalez et al. (2020) and Pfaff et al. (2021) presented a GNN-based architecture based on a system of particles and mesh to simulate a wide range of physical phenomena over time. In particular, both Brandstetter et al. (2022) and Boussif et al. (2022) conducted research with a focus on simulating time-dependent PDEs using graph-based simulations. Sun et al. (2022), similar to our model, combined GNNs and DeepONet to solve power grid transient stability prediction and traffic flow problems. However, unlike the original idea of DeepONet's trunk net, they designed basis functions for the desired time intervals instead of making them align with the spatial basis of the target function domain. As a result, it is more challenging for this model to achieve time extrapolation compared to our model.

## 3 GRAPHDEEPONET FOR TIME-DEPENDENT PDES

### 3.1 PROBLEM STATEMENT

We focus on time-dependent PDEs of the form

$$
\begin{aligned}
\frac{\partial u}{\partial t} &= \mathcal{L}(t, \boldsymbol{x}, u, \frac{\partial u}{\partial x_{(i)}}, \frac{\partial^2 u}{\partial x_{(i)} \partial x_{(j)}}, ...), \\
u(t = 0, \boldsymbol{x}) &= u^0(\boldsymbol{x}), \\
\mathcal{B}[u] &= 0,
\end{aligned}
\tag{1}
$$

where $\mathcal{B}$ is a boundary operator for $t \in \mathbb{R}^+$ and $\boldsymbol{x} := (x_{(1)}, ..., x_{(d)}) \in \Omega \subset \mathbb{R}^d$ with a bounded domain $\Omega$. Denote the frame of solution as $u^k(\boldsymbol{x}) := u(k\Delta t, \boldsymbol{x})$ for a fixed $\Delta t$. Assume that the one solution trajectory for (1) consists of $K_{\text{frame}}$ frames $u^0, ..., u^{K_{\text{frame}}}$. Assume that we have $N_{\text{train}}$ trajectories for the training dataset and $N_{\text{test}}$ trajectories for the test dataset for each different initial condition $u^0(\boldsymbol{x})$. The aim of operator learning for time-dependent PDEs is to learn a mapping $\mathcal{G}^{(k)} : u^0 \mapsto u^k$ from initial condition $u^0(\boldsymbol{x})$ to the solution $u^k(\boldsymbol{x})$ at arbitrary time $k = 1, ..., K_{\text{frame}}$. Denote the approximated solution for $u^k(\boldsymbol{x})$ as $\widetilde{u}^k(\boldsymbol{x})$.

## 3.2 DeepONet for time-dependent PDEs

DeepONet (Lu et al., 2021) is an operator learning model based on the universality of operators from Chen & Chen (1995). It consists of two networks: a branch net $\boldsymbol{\nu}$ and trunk net $\boldsymbol{\tau}$, as illustrated in Figure 1. Firstly, the branch net takes the input function $u^0(\boldsymbol{x})$ as the discretized values at some fixed sensor points. More precisely, it makes the finite dimensional vector $\bar{u}^0$, where $\bar{u}^0 := [u^0(\boldsymbol{x}_0), ..., u^0(\boldsymbol{x}_{N-1})] \in \mathbb{R}^N$ at some fixed sensor points $\boldsymbol{x}_i \in \Omega \subset \mathbb{R}^d$ ($0 \le i \le N-1$). The branch net then takes the input $\bar{u}^0$ and makes the $p-$length vector output $\boldsymbol{\nu}[\bar{u}^0] = [\nu_1[\bar{u}^0], ..., \nu_p[\bar{u}^0]]$. To simulate the time-dependent PDE (1) using DeepONet, we need to handle the two domain variables, $t$ and $\boldsymbol{x}$, of the target function $u(k\Delta t, \boldsymbol{x})$ as inputs to the trunk net. The trunk net takes variables $t$ and $\boldsymbol{x}$ as input and makes the output $\boldsymbol{\tau}(k\Delta t, \boldsymbol{x}) = [\tau_1(k\Delta t, \boldsymbol{x}), ..., \tau_p(k\Delta t, \boldsymbol{x})]$. The outputs of these networks can be regarded as coefficients and basis functions of the target function, and the inner product of the outputs of these two networks approximates the desired target function. The trunk net learns the basis functions of the target function domain separately, which gives it a significant advantage in predicting the values of the target function on an arbitrary grid within its domain. Hence, the ultimate result of DeepONet, denoted as $\widetilde{u}^k(\boldsymbol{x})$, approximates the solution $u^k(\boldsymbol{x})$ by utilizing the branch and trunk nets as follows:

$$\widetilde{u}^k(\boldsymbol{x}) = \sum_{j=1}^p \nu_j[\bar{u}^0]\tau_j(k\Delta t, \boldsymbol{x}), \tag{2}$$

for $k = 1, ..., K_{\text{frame}}$. However, this approach becomes somewhat awkward from the perspective of the Galerkin projection method. Many studies (Hadorn, 2022; Lee et al., 2023; Lu et al., 2022; Meuris et al., 2023) interpret the roles of the branch and trunk nets in DeepONet from the perspective of the basis functions of the target function. The trunk net generates the $p$-basis functions for the target function, while the the branch net generates the $p$-coefficients corresponding to the $p$-basis functions. These coefficients determine the target function, which varies depending on the input function $u^0(\boldsymbol{x})$. Therefore, the trunk net must to produce spatial basis functions in the target function domain that depend only on $\boldsymbol{x}$, but DeepONet (2) also handles the time variable $t$ as an input to the trunk net, which is unnatural. From this perspective, we consider the improved branch net to deal with the time variable $t$ as follows:

$$\widetilde{u}^k(\boldsymbol{x}) = \sum_{j=1}^p \nu_j[\bar{u}^0, k\Delta t]\tau_j(\boldsymbol{x}), \tag{3}$$

so that the coefficient $\nu_j[\bar{u}^0, k\Delta t]$ is dependent on time variable $t$ to show the global behavior of the function. To express the change of the solution over time using the branch net, a more refined model is required. This is the main motivation behind our development of GraphDeepONet $\mathcal{G}_{\text{GDON}}$ : $[\bar{u}^0, k\Delta t] \mapsto u^k$ using a GNN. In the following section, we provide a more detailed explanation of the proposed model, utilizing a GNN to describe how the branch net $\boldsymbol{\nu}$ is constructed and how it incorporates the evolution over time.

## 3.3 Proposed model: GraphDeepONet

For a fixed set of positional sensors $\boldsymbol{x}_i$ ($0 \le i \le N-1$), we formulate a graph $G = (\mathcal{V}, \mathcal{E})$, where each node $i$ belongs to $\mathcal{V}$ and each edge $(i, j)$ to $\mathcal{E}$. The nodes represent grid cells, and the edges signify local neighborhoods. Edges are constructed based on the proximity of node positions, connecting nodes within a specified distance based on the $k$-NN algorithm (See Appendix C.1).

### 3.3.1 Encoder-Processor-Decoder framework

The GraphDeepONet architecture follows an Encode-Process-Decode paradigm similar to Battaglia et al. (2018); Sanchez-Gonzalez et al. (2020); Brandstetter et al. (2022). However, due to our incorporation of the DeepONet structure, the processor includes the gathering of information from all locations, and the decoder involves a reconstruction that enables predictions from arbitrary positions.

**Encoder $\epsilon$.** The encoder maps node embeddings from the function space to the latent space. For a given node $i$, it maps the last solution values at node position $\boldsymbol{x}_i$, denoted as $u_i^0 := u^0(\boldsymbol{x}_i)$, to

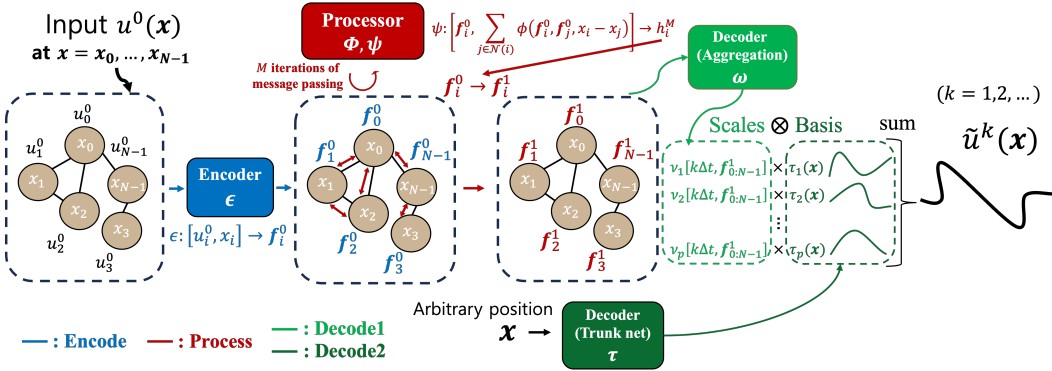

Figure 2: Framework of the proposed GraphDeepONet

the latent embedding vector. Formally, the encoding function $\epsilon : \mathbb{R}^{1+d} \to \mathbb{R}^{d_{\text{lat}}}$ produces the node embedding vector $\boldsymbol{f}_i^0$ as follows:

$$\boldsymbol{f}_i^0 := \epsilon \left( u_i^0, \boldsymbol{x}_i \right) \in \mathbb{R}^{d_{\text{lat}}}, \tag{4}$$

where $\epsilon$ is multilayer perceptron (MLP). It is noteworthy that the sampling method, which includes both the number of sensors $N$ and their respective locations $\boldsymbol{x}_i$ for $0 \le i \le N-1$, can differ for each input.

**Processor $\phi, \psi$.** The processor approximates the dynamic solution of PDEs by performing $M$ iterations of learned message passing, yielding intermediate graph representations. The update equations are given by

$$\boldsymbol{m}_{ij}^m = \phi(\boldsymbol{h}_i^m, \boldsymbol{h}_j^m, \boldsymbol{x}_i - \boldsymbol{x}_j), \tag{5}$$

$$\boldsymbol{h}_i^{m+1} = \psi \left( \boldsymbol{h}_i^m, \sum_{j \in \mathcal{N}(i)} \boldsymbol{m}_{ij}^m \right), \tag{6}$$

for $m = 0, 1, ..., M-1$ with $\boldsymbol{h}_i^0 = \boldsymbol{f}_i^0$, where $\mathcal{N}(i)$ denotes the neighboring nodes of node $i$. Both $\phi$ and $\psi$ are implemented as MLPs. The use of relative positions, i.e., $\boldsymbol{x}_j - \boldsymbol{x}_i$, capitalizes on the translational symmetry inherent in the considered PDEs. After the $M$ iterations of message passing, the processor emits a vector $\boldsymbol{h}_i^M$ for each node $i$. This is used to update the latent vector $\boldsymbol{f}_i^0$ as follows:

$$\boldsymbol{f}_i^1 = \boldsymbol{f}_i^0 + \boldsymbol{h}_i^M, \quad 0 \le i \le N-1. \tag{7}$$

The updated latent vector $\boldsymbol{f}_{0:N-1}^1 := \{\boldsymbol{f}_i^1\}_{i=0}^{N-1}$ is used to predict the next time step solution $u^1(\boldsymbol{x})$.

**Decoder1 - Soft attention aggregation $\omega$.** We first predict the $p-$coefficients for each next timestep. Here, we use the soft attention aggregation layer with the feature-level gating described by Li et al. (2019). The soft attention aggregation $\boldsymbol{\nu} : \mathbb{R}^{d_{\text{lat}} \times N} \to \mathbb{R}^p$ consists of two neural networks to calculate the attention scores and latent vectors as follows:

$$\boldsymbol{\nu}[\boldsymbol{f}_{0:N-1}^1, \Delta t] := \sum_{i=0}^{N-1} \overbrace{\frac{\exp\left(\omega_{\text{gate}}(\boldsymbol{x}_i, \boldsymbol{f}_i^1)/\sqrt{d_{\text{lat}}}\right)}{\sum_{j=0}^{N-1} \exp\left(\omega_{\text{gate}}(\boldsymbol{x}_j, \boldsymbol{f}_j^1)/\sqrt{d_{\text{lat}}}\right)}}^{\text{attention score}} \odot \omega_{\text{feature}}(\Delta t, \boldsymbol{f}_i^1), \tag{8}$$

where $\odot$ represents the element-wise product, and $\omega_{\text{gate}} : \mathbb{R}^{d_{\text{lat}}+d} \to \mathbb{R}^p$ and $\omega_{\text{feature}} : \mathbb{R}^{d_{\text{lat}}+1} \to \mathbb{R}^p$ are MLPs. Note that $\boldsymbol{\nu}$ is well-defined for any number of sensors $N \in \mathbb{N}$. The proposed decoder will be proved to be expressive in Appendix B.

**Decoder2 - Inner product of coefficients and basis $\tau$.** The final output is reconstructed using the $p$−coefficients $\boldsymbol{\nu}[\boldsymbol{f}_{0:N-1}^1, \Delta t]$ and trained global basis via trunk net $\boldsymbol{\tau}(\boldsymbol{x}) = [\tau_1(\boldsymbol{x}), ..., \tau_p(\boldsymbol{x})]$ with $\tau_j : \mathbb{R}^d \to \mathbb{R}$. The next timestep is predicted as

$$\widetilde{u}^1(\boldsymbol{x}) = \sum_{j=1}^{p} \nu_j[\boldsymbol{f}_{0:N-1}^1, \Delta t]\tau_j(\boldsymbol{x}), \tag{9}$$

where $\boldsymbol{\nu}[\boldsymbol{f}_{0:N-1}^1, \Delta t] := [\nu_1, \nu_2, ..., \nu_p] \in \mathbb{R}^p$. The GraphDeepONet is trained using the mean square error $\text{Loss}^{(1)} = \text{MSE}(\widetilde{u}^1(\boldsymbol{x}), u^1(\boldsymbol{x}))$. Since the GraphDeepONet use the trunk net to learn the global basis, it offers a significant advantage in enforcing the boundary condition $\mathcal{B}[u] = 0$ as hard constraints. The GraphDeepONet can enforce periodic boundaries, unlike other graph-based methods, which often struggle to ensure such precise boundary conditions (See Appendix C.6).

### 3.4 Recursive time prediction in latent space

We described the steps of GraphDeepONet from the input function $u^0(x)$ obtained through the encoding-processing-decoding steps to predict the solution at the next timestep, $\widetilde{u}^1(\boldsymbol{x})$. Similar to MP-PDE or MAgNet, by using the predicted $\widetilde{u}^1(\boldsymbol{x})$ as input and repeating the aforementioned steps, we can obtain solutions $\widetilde{u}^k(\boldsymbol{x})$ for $k = 2, 3..., K_{\text{frame}}$. However, rather than recursively evolving in time using the predicted solutions, we propose a method where we evolve in time directly from the encoded latent representation without the need for an additional encoding step. By substituting the value of $\boldsymbol{h}_i^{2M}$ with $\boldsymbol{f}_i^0$ and executing the processor step again, we can derive the second latent vector using the relation

$$\boldsymbol{f}_i^2 = \boldsymbol{f}_i^0 + \boldsymbol{h}_i^{2M}, \quad 0 \leq i \leq N - 1.$$

Employing the $\boldsymbol{f}_i^2$ vectors to produce the function $\widetilde{u}^2(\boldsymbol{x})$, the decoder step remains analogous by using $2\Delta t$ instead of using $\Delta t$. Finally, employing the predicted $\{\widetilde{u}^k(\boldsymbol{x})\}_{k=1}^{K_{\text{frame}}}$ from $u^0(\boldsymbol{x})$ in this manner, we train the GraphDeepONet with $\text{Loss}^{\text{Total}} = \frac{1}{K_{\text{frame}}} \sum_{k=1}^{K_{\text{frame}}} \text{MSE}(\widetilde{u}^k(\boldsymbol{x}), u^k(\boldsymbol{x}))$. The approach of iteratively performing temporal updates to gradually compute solutions is referred to as the **autoregressive method**, which is applied across various PDE solvers (Li et al., 2020; Brandstetter et al., 2022) to obtain time-dependent solutions. When compared to other methods, we anticipate that evolving in time from the encoded embedding vector may result in reduced cumulative errors over time, in contrast to fully recursive approaches. This, in turn, is expected to yield more precise predictions of the solution operator. For computational efficiency, we also use the temporal bundling method suggested by Brandstetter et al. (2022). We group the entire dataset $K_{\text{frame}} + 1$ into sets of $K$ frames each. Given $K$ initial states at time steps $0, \Delta t, \cdots, (K-1)\Delta t$, the model predict the solution's value at subsequent time steps $K\Delta t, \cdots, (2K-1)\Delta t$ and beyond.

### 3.5 Distinctions between GraphDeepONet and other models

Lu et al. (2022) suggested using various network structures, including GNNs, for the branch net of DeepONet, depending on the problem at hand. From this perspective, employing a GNN in the branch net to handle the input function on an irregular grid seems intuitive. However, in the context of time-dependent PDEs, which is the focus of this study, predicting the solution $u(t, \boldsymbol{x})$ of the target function PDE using both $t$ and $\boldsymbol{x}$ as inputs to the trunk net pre-fixes the time domain during training, making extrapolation impossible for future times. In this regard, GraphDeepONet, which considers time $t$ in the branch net instead of the trunk net and utilizes GNNs, distinguishes itself from traditional DeepONet and its variants.

We also highlight that our methodology allows us to predict the solution $\widetilde{u}^k(\boldsymbol{x})$ for any $\boldsymbol{x} \in \Omega \subset \mathbb{R}^d$. This distinguishes our approach from GNN-based PDE solver architectures (Brandstetter et al., 2022; Boussif et al., 2022), which are limited to inferring values only at specified grid points. This is made possible by leveraging one of the significant advantages of the DeepONet model, which uses a trunk in the encoding process that takes the spatial variable $\boldsymbol{x}$ as input. The trunk net creates the basis for the target function, determining the scale at which the coefficients $\nu_i$ are multiplied by the bases, which enables predictions of the solution $u(t, \boldsymbol{x})$ at arbitrary positions. For instance, when predicting the solution as an output function at different points from the predetermined spatial points of the input function, our approach exhibits significant advantages over other GNN-based PDE solver models.

## 3.6 THEORETICAL ANALYSIS OF GRAPHDEEPONET FOR TIME-DEPENDENT PDES

The universality of DeepONet is already well-established (Chen & Chen, 1995; Lanthaler et al., 2022; Prasthofer et al., 2022). However, existing theories that consider time-dependent PDEs tend to focus solely on the solution operator for the target function $u(t, \boldsymbol{x})$ as $\mathcal{G} : u(0, \cdot) \mapsto u(t = T, \cdot)$, or they restrict their considerations to predefined bounded domains of time, such as $\mathcal{G} : u(0, \cdot) \mapsto u(t, \boldsymbol{x})|_{t \in [0,T]}$. In contrast, we employ GNNs to evolve coefficients over time, enabling us to approximate the mapping $\mathcal{G}^{(k)} : u^0 \mapsto u^k$ for arbitrary points in the discretized time domain ($k = 1, 2, ..., K_{\text{frame}}$). Based on the theories from Lanthaler et al. (2022), we aim to provide a theoretical analysis of our GraphDeepONet $\mathcal{G}_{\text{GDON}} : [\bar{u}^0, k\Delta t] \mapsto u^k$, where $\bar{u}^0 = [u^0(\boldsymbol{x}_0), ..., u^0(\boldsymbol{x}_{N-1})] \in \mathbb{R}^N$ at sensor points $\boldsymbol{x}_i \in \Omega \subset \mathbb{R}^d$ ($0 \le i \le N - 1$). The proposed model is capable of effectively approximating the operator, irrespective of the grid's configuration. Our theorem asserts that our model can make accurate predictions at multiple time steps, regardless of the grid's arrangement. The key motivation behind the proof and the significant departure from existing theories lies in the utilization of the representation capability of GNNs for permutation-equivariant functions.

**Theorem 1.** *(**Universality of GraphDeepONet**) Let $\mathcal{G}^{(k)} : H^s(\mathbb{T}^d) \to H^s(\mathbb{T}^d)$ be a Lipschitz continuous operator for $k = 1, ..., K_{\text{frame}}$. Assume that $\mu$ is a probability measure on $L^2(\mathbb{T}^d)$ with a covariance operator characterized by a bounded eigenbasis with eigenvectors $\{\lambda_j\}_{j \in \mathbb{N}}$. Let fixed sensor points $\boldsymbol{x}_i$ ($0 \le i \le N - 1$) be independently selected based on a uniform distribution over $\mathbb{T}^d$ and $\bar{u}^0 = [u^0(\boldsymbol{x}_0), ..., u^0(\boldsymbol{x}_{N-1})]$. If $N$ is sufficiently large, then there exists a GraphDeepONet $\mathcal{G}_{\text{GDON}} : \mathbb{R}^{N+1} \to H^s(\mathbb{T}^d)$ with p-trunk net ensuring that the following holds with probability 1:*

$$\sum_{k=1}^{K_{\text{frame}}} \left\| \mathcal{G}^{(k)}(u^0) - \mathcal{G}_{\text{GDON}}(\bar{u}^0, k\Delta t) \right\|_{L^2(\mu)} \le C \left( \sum_{j > N/C \log(N)} \lambda_j \right)^{1/2} + C p^{-s/d}, \quad (10)$$

*for every $p \in \mathbb{N}$, where the constant $C > 0$ is a function of $N$, $K_{\text{frame}}$, $\mathcal{G}$, and $\mu$.*

There exist graph-based models, including MP-PDE and MAgNet, that attempt to learn the operator $\mathcal{G}^{(1)} : u^0 \mapsto u^1$. These models consider fixed grids for the input $u^0(\boldsymbol{x})$ as nodes and employ message-passing neural networks to transmit information at each node. Consequently, when approximating the target function $u^1(\boldsymbol{x})$, they can only predict function values on specific grids, which can lead to failures in learning the operator $\mathcal{G}^{(1)}$ in certain grid locations. The following theorem provides an error analysis of the learned operator $\mathcal{G}_{\text{graph}}$ based on a GNN, which is only predictable on the same fixed grids. Note that GraphDeepONet is also a graph-based model that uses message-passing neural networks, but it differs in that it utilizes a trunk net $\boldsymbol{\tau}$ based on DeepONet, allowing it to make predictions for all grids.

**Theorem 2.** *(**Failure to learn operator using other graph-based models**) Assume that the sensor points $\boldsymbol{x}_i$ ($0 \le i \le N - 1$) are independently selected following a uniform distribution on $\mathbb{T}^d$. Then, there exists a Lipshitz continuous operator $\mathcal{G}^{(1)} : H^s(\mathbb{T}^d) \to H^s(\mathbb{T}^d)$ such that the following inequality holds with a non-zero probability $\delta > 0$:*

$$\left\| \mathcal{G}^{(1)}(u_0) - \mathcal{G}_{\text{graph}}(u_0) \right\|_{L^2(\mu)} \ge 1/2.$$

We provide proof of this theorem through the construction of a rotational mapping in a periodic domain. The rotation speed increases with distance from the domain's center. Conventional message-passing neural networks, which rely on consistent relative distances for information exchange, are incapable of learning this mapping due to the inherent periodicity; identical inputs at different grid points would erroneously yield the same output. Our approach deviates from this limitation by learning a global latent basis that facilitates the generation of distinct values for identical inputs across various positions. A more detailed proof and discussion are presented in Appendix B.

## 4 EXPERIMENTS

We conduct experiments comparing the proposed GraphDeepONet model with other benchmark models. Firstly, we explore the simulation of time-dependent PDEs by comparing the original DeepONet and VIDON with GraphDeepONet for regular and irregular sensor points. Specifically, we

Table 1: Mean Rel. $L^2$ test errors with standard deviations for 3 types of Burgers' equation dataset using regular/irregular sensor points. Three training trials are performed independently.

| Type of sensor points | Data | FNO-2D | DeepONet variants | | Graph-based model | | GraphDeepONet (Ours) |
|---|---|---|---|---|---|---|---|
| | | | DeepONet | VIDON | MP-PDE | MAgNet | |
| Regular | E1 | **0.1437**$_{\pm 0.0109}$ | 0.3712$_{\pm 0.0094}$ | 0.3471$_{\pm 0.0221}$ | 0.3598$_{\pm 0.0019}$ | 0.2399$_{\pm 0.0623}$ | 0.1574$_{\pm 0.0104}$ |
| | E2 | **0.1343**$_{\pm 0.0108}$ | 0.3688$_{\pm 0.0204}$ | 0.3067$_{\pm 0.0520}$ | 0.2622$_{\pm 0.0019}$ | 0.2348$_{\pm 0.0153}$ | 0.1716$_{\pm 0.0350}$ |
| | E3 | **0.1551**$_{\pm 0.0014}$ | 0.2983$_{\pm 0.0050}$ | 0.2691$_{\pm 0.0145}$ | 0.3548$_{\pm 0.0171}$ | 0.2723$_{\pm 0.0628}$ | 0.2199$_{\pm 0.0069}$ |
| Irregular | E1 | - | 0.3564$_{\pm 0.0467}$ | 0.3430$_{\pm 0.0492}$ | 0.2182$_{\pm 0.0108}$ | 0.4106$_{\pm 0.0864}$ | **0.1641**$_{\pm 0.0006}$ |

assess how well GraphDeepONet predicts in arbitrary position, especially concerning irregular sensor points, compared to models such as MP-PDE, and MAgNet. Furthermore, we include FNO-2D, a well-established model known for operator learning, in our benchmark comparisons. Given the difficulty FNO faces in handling input functions with irregular sensor points, we limited our comparisons to input functions with regular sensor points. We consider the 1D Burgers' equation data from Brandstetter et al. (2022), the 2D shallow water equation data from Takamoto et al. (2022), and the 2D Navier-Stokes (N-S) equation data from Kovachki et al. (2021b). For datasets with periodic boundaries, the GraphDeepONet leveraged the advantage of enforcing the condition (See Appendix C.6). The PyTorch Geometric library (Fey & Lenssen, 2019) is used for all experiments. The relative $L^2$ error by averaging the prediction solutions for all time is used for error estimate. See Appendix C for more details.

**Comparison with DeepONet and its variants**
The fourth and fifth columns in Table 1 display the training results for DeepONet and VIDON, respectively. The DeepONet and VIDON struggled to accurately predict the solutions of Burgers's equation. This is because DeepONet and VIDON lack universal methods to simultaneously handle input and output at multiple timesteps. Figure 3 compares the time extrapolation capabilities of existing DeepONet models. To observe extrapolation, we trained our models using data from time $T_{\text{train}} = [0, 2]$, with inputs ranging from 0 to 0.4, allowing them to predict values from 0.4 to 2. Subsequently, we evaluated the performance of DeepONet, VIDON, and our GraphDeepONet by predicting data $T_{\text{extra}} = [2, 4]$, a range on which they had not been previously trained. Our model clearly demonstrates superior prediction performance when compared to VIDON and Deep-ONet. In contrast to DeepONet and VIDON, which tend to maintain the solutions within the previously

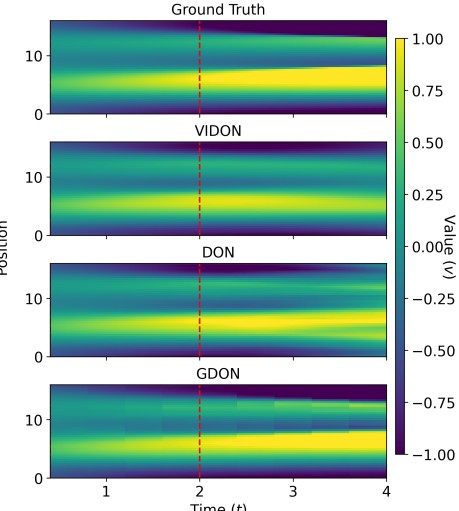

Figure 3: Solution profile in Burgers' equation for time extrapolation simulation using DeepONet, VIDON, and GraphDeepONet.

learned domain $T_{\text{train}}$, the GraphDeepONet effectively learns the variations in the PDE solutions over time, making it more proficient in predicting outcomes for time extrapolation.

**Comparison with GNN-based PDE-solvers** The third, sixth, and seventh columns of Table 1 depict the accuracy of the FNO-2D and GNN-based models. While FNO outperformed the other models on a regular grid, unlike graph-based methods and our approach, it is not applicable to irregular sensor points, which is specifically designed for uniform grids. The F-FNO (Tran et al., 2023), which extends FNO to irregular grids, also faces challenges when applied to the irregular grid of the N-S data (See Appendix C.3). When compared to GNN-based models, with the exception of FNO, our model slightly outperformed MP-PDE and MAgNet, even on an irregular grid. Table 2 summarizes the results of our model along with other graph-based models, including MP-PDE and MAgNet, when applied to various irregular grids for 2D shallow water equation and 2D N-S equation, namely, Irregular I,II, and III for each equation. Remarkably, on one specific grid, MP-PDE outperformed our model. However, the MP-PDE has a significant inconsistency in the predicted performance. In contrast, our model consistently demonstrated high predictive accuracy across all grid cases. This is because, unlike other methods, the solution obtained through GraphDeepONet is continuous in the spatial domain. Figure 4 displays the time-evolution predictions of models trained

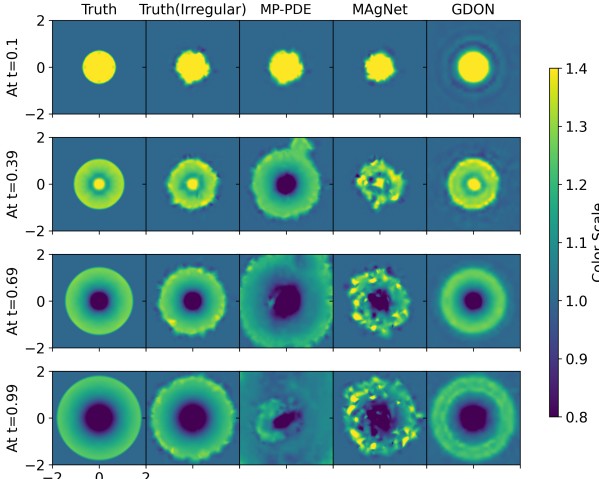

Figure 4: Prediction of 2D shallow water equations on irregular sensor points with distinct training sensor points using graph-based models and GraphDeepONet. The Truth (irregular), MP-PDE, and MAgNet plot the solutions through interpolation using values from the irregular sensor points used during training, whereas GraphDeepONet predicts solutions for all grids directly.

Table 2: Mean Rel. $L^2$ test errors for 2D shallow water equation data using regular/irregular sensor points.

| Data | Type of sensor points | FNO-2D | MP-PDE | MAgNet | GraphDeepONet (Ours) |
|---|---|---|---|---|---|
| 2D shallow | Regular | 0.0025 | **0.0014** | 0.0078 | 0.0124 |
| | Irregular I | - | 0.2083 | 0.0662 | **0.0227** |
| | Irregular II | - | 0.2154 | 0.0614 | **0.0287** |
| | Irregular III | - | **0.0140** | 0.0919 | 0.0239 |
| 2D N-S | Regular | **0.0395** | 0.5118 | 0.3653 | 0.1287 |
| | Irregular I | - | 1.1176 | 0.4564 | **0.1243** |
| | Irregular II | - | **0.1207** | 0.4676 | 0.1257 |
| | Irregular III | - | **0.1240** | 0.4306 | 0.1289 |

on the shallow water equation for an initial condition. The GNN-based models are trained on fixed irregular sensors as seen in the second column and are only capable of predicting on the same grid, necessitating interpolation for prediction. In contrast, GraphDeepONet leverages the trunk net, enabling predictions at arbitrary grids, resulting in more accurate predictions.

## 5  CONCLUSION AND DISCUSSION

The proposed GraphDeepONet represents a significant advancement in the realm of PDE solution prediction. Its unique incorporation of time information through a GNN in the branch net allows for precise time extrapolation, a task that has long challenged traditional DeepONet and its variants. Additionally, our method outperforms other graph-based PDE solvers, particularly on irregular grids, providing continuous spatial solutions. Furthermore, GraphDeepONet offers theoretical assurance, demonstrating its universal capability to approximate continuous operators across arbitrary time intervals. Altogether, these innovations position GraphDeepONet as a powerful and versatile tool for solving PDEs, especially in scenarios involving irregular grids. While our GraphDeepONet model has demonstrated promising results, one notable limitation is its current performance on regular grids, where it is outperformed by FNO. Addressing this performance gap on regular grids remains an area for future improvement. As we have employed the temporal bundling method in our approach, one of our future endeavors includes exploring other techniques utilized in DeepONet-related models and GNN-based PDE solver models to incorporate them into our model. Furthermore, exploring the extension of GraphDeepONet to handle more complex 2D time-dependent PDEs or the N-S equations, could provide valuable insights for future applications.

## 6 REPRODUCIBILITY STATEMENT

The data employed in this study consists of Burgers' equation data obtained from Brandstetter et al. (2022); Boussif et al. (2022) and 2D shallow water equation data from Takamoto et al. (2022) as described in the main text. Experiment details can be found in both the main text and the Appendix C. For the sake of reproducibility, we submit the code for the fundamental settings as supplemental material.

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

# A  NOTATIONS

The notations in the paper is summarized in Table 3.

Table 3: Notations

| Notation | Meaning |
|---|---|
| $t$ | the spatial variable |
| $d$ | the dimension of spatial domain |
| $\boldsymbol{x}$ | the spatial variable in $d$ dimension |
| $\boldsymbol{x}_i$ $(i = 0, 1, ..., N-1)$ | the $N$-fixed sensor point in the spatial domain |
| $\Delta t$ | the discretized time |
| $K_{\text{frame}} + 1$ | the number of frames in one solution trajectory |
| $K$ | the number of grouping frames for temporal bundling method |
| $u^k(\boldsymbol{x})$ $(k = 0, 1, ..., K_{\text{frame}})$ | the solution at time $t = k\Delta t$ |
| $\bar{u}^k(\boldsymbol{x})$ $(k = 0, 1, ..., K_{\text{frame}})$ | the values of solution at time $t = k\Delta t$ in fixed sensor points |
| $\widetilde{u}^k(\boldsymbol{x})$ $(k = 0, 1, ..., K_{\text{frame}})$ | the approximated solution at time $t = k\Delta t$ |
| $\mathcal{G}^{(k)}$ | the operator from the initial condition to the solution at time $k\Delta t$ |
| $\mathcal{G}_{\text{GDON}}$ | the approximated operator using GraphDeepONet |
| $\mathcal{G}_{\text{graph}}$ | the approximated operator using other graph-based PDE solver |
| $p$ | the number of basis (or coefficients) in DeepONet |
| $\boldsymbol{\nu}$ | the branch net (or decoder) in DeepONet (or GraphDeepONet) |
| $\boldsymbol{\tau}$ | the trunk net in DeepONet (or GraphDeepONet) |
| $\boldsymbol{\epsilon}$ | the encoder in GraphDeepONet |
| $\boldsymbol{\phi}, \boldsymbol{\psi}$ | the neural networks of processor in GraphDeepONet |
| $\boldsymbol{\omega}$ | the neural network of decoder in GraphDeepONet |
| $\boldsymbol{f}_i$ $(i = 0, 1, ..., N-1)$ | the feature vector at node $i$ |

# B  DETAILS ON THE UNIVERSALITY OF THE PROPOSED GRAPHDEEPONET

We first define the Lipschitz continuity of the operator.

**Definition 1.** *For $\alpha > 0$, an operator $\mathcal{G} : X \to Y$ is **Lipschitz continuous** if there exists a constant $C_{\mathcal{G}}$ such that*

$$\|\mathcal{G}(f_1) - \mathcal{G}(f_2)\|_Y \le C_{\mathcal{G}}\|f_1 - f_2\|_X, \quad \forall f_1, f_2 \in X.$$

Note that if there exists a constant $C_{\mathcal{G}}$ such that $\mathcal{G} : X(\subset Z) \to X$ such that $\|\mathcal{G}(f_1) - \mathcal{G}(f_2)\|_X \le C_{\mathcal{G}}\|f_1 - f_2\|_X$ we say that $\mathcal{G}$ is Lipschitz continuous in $Z$.

Throughout the theorem in this section, we consider the Lipschitz operator $\mathcal{G}^{(k)} : \mathbb{H}^s(\mathbb{T}^d) \to \mathbb{H}^s(\mathbb{T}^d)$ and probability measure $\mu$ which satisfy the following condition.

**Assumption 1.** *There exists a constant $M$ such that the following inequality holds for any $k \in \{1, \ldots, K_{\text{frame}}\}$.*

$$\int_{H^s(\mathbb{T}^d)} \|\mathcal{G}^{(k)}(u)\|^2 d\mu(u) \le M.$$

**Assumption 2.** *The covariance operator $\Gamma^k = \int_{L^2(\mathbb{T}^d)} u \otimes u \, d\mathcal{G}_*^{(k)}\mu$ has bounded eigenbasis for any $k \in 1, \ldots, K_{\text{frame}}$ where $d\mathcal{G}_*^{(k)}\mu$ is push forward measure of $\mu$.*

Figure 5 shows the pictorial description of the proposed GraphDeepONet model. The real line denotes the computational process by GraphDeepONet and the dashed line denotes the process through the correct operator $\mathcal{G}$.

The first encoding step $\mathcal{E}_1 : L^2(\mathbb{T}^d) \to \{\mathbb{R} \cup \mathbb{T}^d\}^N$ maps the initial condition $u^0(\boldsymbol{x})$ to $\{(u^0(\boldsymbol{x}_i), \boldsymbol{x}_i)\}_{i=1}^N$ which denote finite observations at $N$ sensor points. The second encoding step $\mathcal{E}_2 : \{\mathbb{R} \cup \mathbb{T}^d\}^N \to \{\mathbb{R}^{d_{\text{lat}}} \cup \mathbb{T}^d\}^N$ is constructed by fully connected neural network which embeds

the function value $u^0(\boldsymbol{x}_i)$ at each location into the embedding space whose dimension is $d_{\text{lat}}$. For $\epsilon$ in Equation (4), it can be written as

$$\mathcal{E}_2(\{u^0(\boldsymbol{x}_i), \boldsymbol{x}_i\}_{i=1}^N) := \{\epsilon(u^0(\boldsymbol{x}_i), \boldsymbol{x}_i), \boldsymbol{x}_i\}_{i=1}^N.$$

The processor $\mathcal{P} : \{\mathbb{R}^{d_{\text{lat}}} \cup \mathbb{T}^d\}^N \to \{\mathbb{R}^{d_{\text{lat}}} \cup \mathbb{T}^d\}^N$ updates the value at each node by message passing algorithm by Equation (5) and (6). The first decoder $\mathcal{D}_1 : \{\mathbb{R}^{d_{\text{lat}}} \cup \mathbb{T}^d\}^N \to \mathbb{R}^p$ in Equation (8) obtain the predicted coefficient of function in $L^2(\mathbb{T}^d)$ at each time from the given all values in latent space. The second decoder $\mathcal{D}_2 : \mathbb{R}^p \to L^2(\mathbb{T}^d)$ in Equation (9) finally predicts the function using the learned basis function $\{\tau_i(\boldsymbol{x})\}_{i=1}^p$ in $L^2(\mathbb{T}^d)$ at each time.

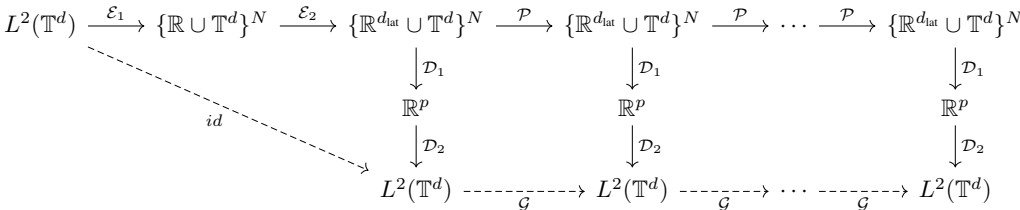

Figure 5: Diagram of the GraphDeepONet

Now for given initial condition $u_0(\boldsymbol{x})$, we denote the approximated solution at time $k\Delta t$ by $\mathcal{N}_k : L^2(\mathbb{T}^d) \to L^2(\mathbb{T}^d)$ is defined as $\mathcal{N}_k := \mathcal{D}_2 \circ \mathcal{D}_1 \circ \mathcal{P}^{(k)} \circ \mathcal{E}_2 \circ \mathcal{E}_1$. Let us denote the distribution of input function $u_0$ by $\mu$. Then, the $L^2$ error of our approximation $\mathcal{G}_{\text{DON}}$ to operator $\mathcal{G}$ is expressed by

$$\hat{\mathcal{E}} := \sum_{k=1}^{K_{\text{frame}}} \left\| \mathcal{G}^{(k)}(u^0) - \mathcal{G}_{\text{GDON}}(\bar{u}^0, k\Delta t) \right\|_{L^2(\mu)}$$

$$= \sum_{k=1}^{K_{\text{frame}}} \int_{H^s(\mathbb{T}^d)} \int_{\mathbb{T}^d} |\mathcal{G}^{(k)}(u)(\boldsymbol{x}) - \mathcal{N}_k(u)(\boldsymbol{x})|^2 d\boldsymbol{x} d\mu(u). \quad (11)$$

Now, we can come up with a pseudo-inverse of $\mathcal{E}_1$ and $\mathcal{D}_2$ such that $\mathcal{D}_2^+ \circ \mathcal{D}_2 = id$ and $\mathcal{E}_1 \circ \mathcal{E}_1^+ = id$. For example, the image of $\mathcal{E}_1^+$ consists of a piecewise linear function that connects the given input points $\{(u^0(\boldsymbol{x}_i), \boldsymbol{x}_i)\}_{i=0}^{N-1}$ and $\mathcal{D}_2^+ : L^2(\mathbb{T}^d) \to \mathbb{R}^p, f \mapsto (\langle f, \tau_1 \rangle, \ldots, \langle f, \tau_p \rangle)$. Then, we can obtain the following inequality by the Lipschitz continuity of the operator.

**Lemma 1.** *Suppose that $\mathcal{D}_2, \mathcal{D}_2^+$ are Lipschitz continuous and $\mathcal{G} : H^s(\mathbb{T}^d) \to H^s(\mathbb{T}^d)$ is Lipschitz continuous in $L^2(\mathbb{T}^d)$. For the error $\hat{\mathcal{E}}$ defined in* (11)*, the following inequality holds*

$$\hat{\mathcal{E}} \leq \sum_{k=1}^{K_{frame}} (C_{\mathcal{D}_2} \int_{L^2(\mathbb{T}^d)} \int_{\{\mathbb{R} \cup \mathbb{T}^d\}^N} |D_1 \circ \mathcal{P}^{(k)} \circ \mathcal{E}_2 - \mathcal{D}_2^+ \circ \mathcal{G}^{(k)} \circ \mathcal{E}_1^+|^2 d\boldsymbol{y} d(\mathcal{E}_{1*}\mu)$$

$$+ C_{\mathcal{D}_2 \circ \mathcal{D}_2^+} C_{\mathcal{G}}^k \int_{L^2(\mathbb{T}^d)} (\int_{\mathbb{T}^d} |\mathcal{E}_1^+ \circ \mathcal{E}_1 - id|^2 d\boldsymbol{x}) d\mu + \int_{L^2(\mathbb{T}^d)} \int_{\mathbb{T}^d} |\mathcal{D}_2 \circ \mathcal{D}_2^+ - id|^2 d(\mathcal{G}_*^{(k)}\mu)),$$

*where $\mathcal{E}_{1*}\mu$ and $\mathcal{G}_*^{(k)}\mu$ are push-forward measure under $\mathcal{E}_1$ and $\mathcal{G}^{(k)}$ respectively.*

*Proof.* First, we decompose $\mathcal{N}_k - \mathcal{G}^{(k)}$ into the following four terms.

$$\mathcal{N}_k - \mathcal{G}^{(k)} = \mathcal{D}_2 \circ \mathcal{D}_1 \circ \mathcal{P}^{(k)} \circ \mathcal{E}_2 \circ \mathcal{E}_1 - \mathcal{G}^{(k)}$$

$$= \mathcal{D}_2 \circ \mathcal{D}_1 \circ \mathcal{P}^{(k)} \circ \mathcal{E}_2 \circ \mathcal{E}_1 - \mathcal{D}_2 \circ \mathcal{D}_2^+ \circ \mathcal{G}^{(k)} \circ \mathcal{E}_1^+ \circ \mathcal{E}_1$$

$$+ \mathcal{D}_2 \circ \mathcal{D}_2^+ \circ \mathcal{G}^{(k)} \circ \mathcal{E}_1^+ \circ \mathcal{E}_1 - \mathcal{D}_2 \circ \mathcal{D}_2^+ \circ \mathcal{G}^{(k)}$$

$$+ \mathcal{D}_2 \circ \mathcal{D}_2^+ \circ \mathcal{G}^{(k)} - \mathcal{G}^{(k)}$$

For the first term, we can derive the following inequality by the Lipschitz continuity of the operator,

$$\int_{L^2(\mathbb{T}^d)} \int_{\mathbb{T}^d} |\mathcal{D}_2 \circ \mathcal{D}_1 \circ \mathcal{P}^{(k)} \circ \mathcal{E}_2 \circ \mathcal{E}_1 - \mathcal{D}_2 \circ \mathcal{D}_2^+ \circ \mathcal{G}^{(k)} \circ \mathcal{E}_1^+ \circ \mathcal{E}_1|^2 d\boldsymbol{x} d\mu$$

$$\leq C_{\mathcal{D}_2} \int_{L^2(\mathbb{T}^d)} \int_{\mathbb{R}^N} |D_1 \circ \mathcal{P}^{(k)} \circ \mathcal{E}_2 - \mathcal{D}_2^+ \circ \mathcal{G}^{(k)} \circ \mathcal{E}_1^+|^2 d\boldsymbol{x} d(\mathcal{E}_{1*}\mu),$$

where $\mathcal{E}_{1*}\mu$ is push-forward measure under $\mathcal{E}_1$.

$$\int_{L^2(\mathbb{T}^d)} \int_{\mathbb{T}^d} |\mathcal{D}_2 \circ \mathcal{D}_2^+ \circ \mathcal{G}^{(k)} \circ \mathcal{E}_1^+ \circ \mathcal{E}_1 - \mathcal{D}_2 \circ \mathcal{D}_2^+ \circ \mathcal{G}^{(k)}|^2 d\boldsymbol{x} d\mu$$

$$\leq C_{\mathcal{D}_2 \circ \mathcal{D}_2^+} \int_{L^2(\mathbb{T}^d)} C_{\mathcal{G}} (\int_{\mathbb{T}^d} |\mathcal{G}^{(k-1)} \circ \mathcal{E}_1^+ \circ \mathcal{E}_1 - \mathcal{G}^{(k-1)}|^2 d\boldsymbol{x}) d\mu$$

$$\dots$$

$$\leq C_{\mathcal{D}_2 \circ \mathcal{D}_2^+} C_{\mathcal{G}}^k \int_{L^2(\mathbb{T}^d)} (\int_{\mathbb{T}^d} |\mathcal{E}_1^+ \circ \mathcal{E}_1 - id|^2 d\boldsymbol{x}) d\mu$$

Finally,

$$\int_{L^2(\mathbb{T}^d)} \int_{\mathbb{T}^d} |\mathcal{D}_2 \circ \mathcal{D}_2^+ \circ \mathcal{G}^{(k)} - \mathcal{G}^{(k)}|^2 d\boldsymbol{x} d\mu$$

$$= \int_{L^2(\mathbb{T}^d)} \int_{\mathbb{T}^d} |\mathcal{D}_2 \circ \mathcal{D}_2^+ - id|^2 d(\mathcal{G}_*^{(k)}\mu)$$

where $\mathcal{G}_*^{(k)}\mu$ is push forward measure under $\mathcal{G}^{(k)}$.

$\square$

We introduce Theorem 3.5 from (Lanthaler et al., 2022) which can handle the error related to the second step of the decoder. It implies that the whole space $H^s(\mathbb{T}^d)$ can be approximated by the truncated Fourier domain with a finite Fourier basis function.

**Theorem 3.** *(Theorem 3.5, (Lanthaler et al., 2022)) Let us consider a operator $\mathcal{G} : H^s(\mathbb{T}^d) \to H^s(\mathbb{T}^d)$ which is Lipschitz in $L^2(\mathbb{T}^d)$. Assume that the distribution corresponding to $H^s(\mathbb{T}^d)$ follows $\mu$ such that*

$$\int_{H^s(\mathbb{T}^d)} \|\mathcal{G}(u)\|^2 d\mu(u) \leq M$$

*there exists a constant $C = C(d, s, M)$ such that for any $p \in \mathbb{N}$, there exists a trunk network $\tau : \mathbb{R}^n \to \mathbb{R}^p$ such that*

$$\int_{\mathbb{T}^d} |\mathcal{D}_2 \circ \mathcal{D}_2^+ - id|^2 d\boldsymbol{x} \leq CP^{-2s/n}$$

*Furthermore, $\mathcal{D}_2$ and $\mathcal{D}_2^+$ are Lipschitz continuous with constants less than 2.*

We introduce Theorem 3.7 from (Lanthaler et al., 2022) which can handle the error related to the first step of the encoding. Note that the following theorem states finite observation points through sampling the sensor on uniform distribution really capture the function which is defined in $\mathbb{T}^d$ given a large number of sensor points.

**Theorem 4.** *(Theorem 3.7, (Lanthaler et al., 2022)) Assume that the eigenbasis of covariance operator $\overline{\Gamma}$ is bounded in $L^\infty$ sense. Suppose that $\{\boldsymbol{x}_i\}_{i=0}^{N-1}$ is sampled from the uniform distribution in $\mathbb{T}^d$. Then, there exists a constant $C = C(\sup_{p \in \mathbb{N}} \|\phi_p\|_{L^\infty(\mathbb{T}^d)}, d)$ and large integer $N_0$ such that the following inequality holds with probability 1 if $N \geq N_0$.*

$$\int_{L^2(\mathbb{T}^d)} |\mathcal{E}_1^+ \circ \mathcal{E}_1 - id|^2 d\mu \leq C \sqrt{\sum_{p > N/C log(N)} \lambda_p}$$

From now, we assume that the second encoder $\mathcal{E}_2$ is identity map (i.e. $d_{\text{lat}} = 1$) for the simplicity. If we address the case when $d_{\text{lat}} > 1$, all of the works can be done with the same proof by setting $\mathcal{E}_2(\{(u^0(\boldsymbol{x}_i), \boldsymbol{x}_i)\}_{i=0}^{N-1}) = \{(u^0(\boldsymbol{x}_i), 0, \ldots, 0, \boldsymbol{x}_i)\}_{i=0}^{N-1}$

**Lemma 2.** *Assume that there exists a constant $U$ such that $|u^0(\boldsymbol{x}_i)| < U$ for all $0 \leq i \leq N - 1$. For any $\epsilon > 0$, there exists neural network $\omega_{gate}$ and $\omega_{feature}$ such that the following holds for any $\{u^0(\boldsymbol{x}_i), \boldsymbol{x}_i\}_{i=0}^{N-1}$.*

$$\|(\mathcal{D}_1 - \mathcal{D}_2^+ \circ \mathcal{E}_1^+)(\{u^0(\boldsymbol{x}_i), \boldsymbol{x}_i\}_{i=0}^{N-1})\|_{l^p} \leq \epsilon$$

*Proof.* Note that we address the optimal basis function as $\tau_j^*$ in Theorem 3 and interpret $\mathcal{D}_2^+ \circ \mathcal{E}_1^+$ as discrete transform. We would like to prove that $\mathcal{D}_1$ is a good approximator of discrete transform. We have a function $u$ defined on a $d$-dimensional grid, where the grid points are given by the vectors $\boldsymbol{x}_0, \boldsymbol{x}_1, \ldots, \boldsymbol{x}_{N-1}$ in $\mathbb{T}^d$, and the corresponding function values are $u^0(\boldsymbol{x}_0), u^0(\boldsymbol{x}_1), \ldots, u^0(\boldsymbol{x}_{N-1})$.

For $\mathcal{D}_2^+ \circ \mathcal{E}_1^+(\{(u^0(\boldsymbol{x}_i), \boldsymbol{x}_i)\}_{i=0}^{N-1})$, the corresponding coefficients $c_j$ for each basis function $\tau_j^*$ can be computed by:

$$c_j = \sum_{i=0}^{N-1} u^0(\boldsymbol{x}_i)\tau_j^*(\boldsymbol{x}_i), \quad \text{for} \quad j \in \mathbb{Z}^d.$$

Without the consideration of softmax related to normalization in (8), we can select the two networks $\omega_{\text{gate}}(u^0(\boldsymbol{x}_i), \boldsymbol{x}_i)$ and $\omega_{\text{feature}}(u^0(\boldsymbol{x}_i), \boldsymbol{x}_i)$ as the close approximator of $u^0(\boldsymbol{x}_i)$ and $\{\tau_j(\boldsymbol{x}_i)\}_{j=1,\ldots,p}$ by the universal approximation theorem on fully connected neural networks in $L^\infty$ sense. Then the classical argument with the triangle inequality and the boundedness of $u^0(\boldsymbol{x}_i)$ shows that the product sum $\mathcal{D}_1(\{u^0(\boldsymbol{x}_i), \boldsymbol{x}_i\}_{i=0}^{N-1}) = \sum_{i=0}^{N-1} \omega_{\text{gate}}(u^0(\boldsymbol{x}_i), \boldsymbol{x}_i)\omega_{\text{feature}}(u^0(\boldsymbol{x}_i), \boldsymbol{x}_i)$ can indeed approximate the $\mathcal{D}_2^+ \circ \mathcal{E}_1^+$. $\square$

**Definition 2.** *A fully interconnected GNN is a recurrent process that produces a series of functions $h^{(m)}$ for $m \geq 0$. Given each node feature value $\{f_i\}_{i=0}^{N-1}$ with the edge feature $w_{i,j} \in \mathbb{R}^1$, the procedure is defined as follows:*

$$\begin{aligned} &\text{for } k = 0 : \; h_i^{(0)} = f_i, \\ &\text{for } k > 0 : \; h_i^{(m)} = \psi_m\left(h_i^{(m-1)}, \sum_{j \in [n]_{-i}} \phi_k(h_j^{(m-1)}, w_{i,j})\right). \end{aligned}$$

Note that this definition considers the case when $\psi_m$ and $\phi_m$ are functions, not neural networks. There is some differences since our $\mathcal{P}$ is indeed constructed by neural network. With this regard, we summarize the theorem 3.4 and corollary 3.14 from (Fereydounian et al., 2022).

Now, we define the permutation-invariance of multidimensional function.

**Definition 3.** *Suppose that a function $f : (\mathbb{R}^{d_{in}})^M \to (\mathbb{R}^{d_{out}})^M$ with $f_i : \mathbb{R}^{d_{in}} \to \mathbb{R}^{d_{out}}$ is defined by*

$$f(\boldsymbol{x}_1, \ldots, \boldsymbol{x}_M) = (f_1(\boldsymbol{x}_1, \ldots, \boldsymbol{x}_M), \ldots, f_M(\boldsymbol{x}_1, \ldots, \boldsymbol{x}_M)).$$

*Then, $f$ is said to be permutation-compatible (equivariant) if, for any bijective function $\pi : [N] \to [N]$, we have*

$$(f_1(\boldsymbol{x}_{\pi(1)}, \ldots, \boldsymbol{x}_{\pi(N)}), \ldots, f_M(\boldsymbol{x}_{\pi(1)}, \ldots, \boldsymbol{x}_{\pi(n)})) = (f_{\pi(1)}(\boldsymbol{x}_1, \ldots, \boldsymbol{x}_N), \ldots, f_{\pi(N)}(\boldsymbol{x}_1, \ldots, \boldsymbol{x}_N)),$$

*where $\boldsymbol{x}_i \in \mathbb{R}^{d_{in}}$ for all $i \in [N]$.*

Note that considering the position with the function value, the process such as $\mathcal{E}_2$ and $\mathcal{P}$ become permutation equivariant.

**Lemma 3.** *Let $\mathcal{F} : \mathcal{Y} \subset \{\mathbb{R} \cup \mathbb{T}^d\}^N \to \{\mathbb{R} \cup \mathbb{T}^d\}^N$ be a permutation compatible and continuous function. Suppose that $\mathcal{Y}$ is a bounded set. For any $\epsilon > 0$, there exists a $\mathcal{P}$ with a finite depth such that*

$$\|\mathcal{P}(y) - \mathcal{F}(y)\|_{l^2(\{\mathbb{R}\cup\mathbb{T}^d\}^N)} \leq \epsilon, \forall y \in \mathcal{Y}$$

*Proof.* Note that the input node feature $\{(u(\boldsymbol{x}_i), \boldsymbol{x}_i)\}_{i=0}^{N-1}$ should be different for every node since there is no duplicated position values $\boldsymbol{x}_i$. By Theorem 3.4 from (Fereydounian et al., 2022), there exists a graph neural network $H$ with finite depth $K$ such that

$$\|H^{(k)}(y) - \mathcal{F}(y)\|_{l^2(\{\mathbb{R} \cup \mathbb{T}^d\}^N)} \leq \epsilon, \forall y \in \mathcal{Y}.$$

By the corollary 3.14 from Fereydounian et al. (2022), $\{\rho_k, \phi_k\}_{k=1}^{K}$ can be selected by continuous functions by the continuity of $\mathcal{F}$. Since $\mathcal{Y}$ is bounded, the universal approximation theorem on fully connected neural networks in $L^\infty$ sense gives the desired results. □

### B.1 PROOF OF THEOREM 1

We now calculate the upper limit of the term on the right-hand side in the inequality of Lemma B. The only challenging component to evaluate on the right-hand side is the term

$$\|\mathcal{D}_1 \circ \mathcal{P}^{(k)} - \mathcal{D}_2^+ \circ \mathcal{G}^{(k)} \circ \mathcal{E}_1^+\|_{l^2(\{\mathbb{R} \cup \mathbb{T}^d\}^N)}.$$

We outline the proof here of how we can estimate the term with the above theorem.

*Proof.* We begin the proof by assuming first that $\mathcal{E}_2$ is an identity map with $d_{\text{lat}} = 1$. Note that the proof is similar when $d_{\text{lat}} \geq 2$ by just concatenating the identity map with 0. By triangle inequality,

$$\|\mathcal{D}_1 \circ \mathcal{P}^{(k)} - \mathcal{D}_2^+ \circ \mathcal{G}^{(k)} \circ \mathcal{E}_1^+\|_{l^2(\{\mathbb{R} \cup \mathbb{T}^d\}^N)}$$
$$\leq \|\mathcal{D}_1 \circ \mathcal{P}^{(k)} - \mathcal{D}_2^+ \circ \mathcal{E}_1^+ \circ \mathcal{P}^{(k)}\|_{l^2(\{\mathbb{R} \cup \mathbb{T}^d\}^N)}$$
$$\quad + \|\mathcal{D}_2^+ \circ \mathcal{E}_1^+ \circ \mathcal{P}^{(k)} - \mathcal{D}_2^+ \circ \mathcal{E}_1^+ \circ (\mathcal{E}_1 \circ \mathcal{G} \circ \mathcal{E}_1^+)^{(k)}\|_{l^2(\{\mathbb{R} \cup \mathbb{T}^d\}^N)}$$
$$\quad + \|\mathcal{D}_2^+ \circ \mathcal{E}_1^+ \circ (\mathcal{E}_1 \circ \mathcal{G} \circ \mathcal{E}_1^+)^{(k)} - \mathcal{D}_2^+ \circ \mathcal{E}_1^+ \circ \mathcal{E}_1 \circ \mathcal{G}^{(k)} \circ \mathcal{E}_1^+\|_{l^2(\{\mathbb{R} \cup \mathbb{T}^d\}^N)}$$
$$\quad + \|\mathcal{D}_2^+ \circ \mathcal{E}_1^+ \circ \mathcal{E}_1 \circ \mathcal{G}^{(k)} \circ \mathcal{E}_1^+ - \mathcal{D}_2^+ \circ \mathcal{G}^{(k)} \circ \mathcal{E}_1^+\|_{l^2(\{\mathbb{R} \cup \mathbb{T}^d\}^N)}$$

The first term on the right side can be arbitrarily small by Lemma 2. For the second term, we note that $\mathcal{E}_1 \circ \mathcal{G} \circ \mathcal{E}_1^+$ is permutation compatible function. Furthermore, $\mathcal{D}_2^+, \mathcal{E}_1$ is continuous with Lipshcitz constant $C_{\mathcal{E}_1}$ and $C_{\mathcal{D}_2^+}$ since the general Sobolev inequality induces the embedding from $H^s(\mathbb{T}^d)$ into the Hölder space $C^{s-[d/2]-1}(\mathbb{T}^d)$ when $s \geq [d/2] + 1$. Therefore,

$$\|\mathcal{D}_2^+ \circ \mathcal{E}_1^+ \circ \mathcal{P}^{(k)} - \mathcal{D}_2^+ \circ \mathcal{E}_1^+ \circ (\mathcal{E}_1 \circ \mathcal{G} \circ \mathcal{E}_1^+)^{(k)}\|_{l^2(\{\mathbb{R} \cup \mathbb{T}^d\}^N)} \leq C_{\mathcal{D}_2^+} C_{\mathcal{E}_1^+} \|\mathcal{P}^k - (\mathcal{E}_1 \circ \mathcal{G} \circ \mathcal{E}_1^+)^{(k)}\|_{l^2(\{\mathbb{R} \cup \mathbb{T}^d\}^N)}.$$

By Lemma 3, there exists a $\mathcal{P}$ such that

$$|\mathcal{P} - \mathcal{E}_1 \circ \mathcal{G} \circ \mathcal{E}_1^+|(y) \leq \epsilon, \forall y \in \mathcal{Y}.$$

$$|\mathcal{P}^{(k)} - (\mathcal{E}_1 \circ \mathcal{G} \circ \mathcal{E}_1^+)^{(k)}|(y) \leq |(\mathcal{P} - (\mathcal{E}_1 \circ \mathcal{G} \circ \mathcal{E}_1^+))^{(k-1)} \circ (\mathcal{P} - \mathcal{E}_1 \circ \mathcal{G} \circ \mathcal{E}_1^+)|(y)$$
$$\leq (C_{\mathcal{P}} + C_{\mathcal{E}_1} C_{\mathcal{G}} \mathcal{E}_2^{k-1})\epsilon.$$

For the third term, the following holds by the Lipshitz continuity

$$\|\mathcal{D}_2^+ \circ \mathcal{E}_1^+ \circ (\mathcal{E}_1 \circ \mathcal{G} \circ \mathcal{E}_1^+)^{(k)} - \mathcal{D}_2^+ \circ \mathcal{E}_1^+ \circ \mathcal{E}_1 \circ \mathcal{G}^{(k)} \circ \mathcal{E}_1^+\|_{l^2(\{\mathbb{R} \cup \mathbb{T}^d\}^N)}$$
$$\leq C_{\mathcal{D}_2^+} C_{\mathcal{E}_1^+} \|(\mathcal{E}_1 \circ \mathcal{G} \circ \mathcal{E}_1^+)^{(k)} - \mathcal{E}_1 \circ \mathcal{G}^{(k)} \circ \mathcal{E}_1^+\|_{l^2(\{\mathbb{R} \cup \mathbb{T}^d\}^N)}$$

If $k = 2$, then we can estimate the upper bound of the term with the following statement.

$$\int_{\{\mathbb{R} \cup \mathbb{T}^d\}^N} |(\mathcal{E}_1 \circ \mathcal{G} \circ \mathcal{E}_1^+)^{(2)} - \mathcal{E}_1 \circ \mathcal{G}^{(2)} \circ \mathcal{E}_1^+|^2 d\mathcal{E}_1^* \mu$$
$$= \int_{L^2(\mathbb{T}^d)} |(\mathcal{E}_1 \circ \mathcal{G} \circ \mathcal{E}_1^+)^{(2)} \circ \mathcal{E}_1 - \mathcal{E}_1 \circ \mathcal{G}^{(2)} \circ \mathcal{E}_1^+ \circ \mathcal{E}_1|^2 d\mu$$
$$\leq \int_{L^2(\mathbb{T}^d)} |\mathcal{E}_1 \circ \mathcal{G} \circ \mathcal{E}_1^+ \circ \mathcal{E}_1 \circ \mathcal{G} - \mathcal{E}_1 \circ \mathcal{G}^{(2)}|^2 d\mu + (C_{\mathcal{E}_1}^2 C_{\mathcal{G}}^2 C_{\mathcal{E}_1^+} + C_{\mathcal{E}_1} C_{\mathcal{G}}^2)^2 \int_{L^2(\mathbb{T}^d)} |id - \mathcal{E}_1^+ \circ \mathcal{E}_1|^2 d\mu$$

By the Lipschitz continuity, we can obtain the following inequality.

$$\int_{L^2(\mathbb{T}^d)} |\mathcal{E}_1 \circ \mathcal{G} \circ \mathcal{E}_1^+ \circ \mathcal{E}_1 \circ \mathcal{G} - \mathcal{E}_1 \circ \mathcal{G}^{(2)}|^2 d\mu$$

$$\leq C_{\mathcal{E}_1}^2 C_{\mathcal{G}}^2 \int_{L^2(\mathbb{T}^d)} |\mathcal{E}_1^+ \circ \mathcal{E}_1 - id|^2 dG_{\#}\mu$$

With the inductive step for $k$ with theorem 5, we can conclude that there exists constant $C$ such that the following inequality holds with probability 1

$$\int_{\{\mathbb{R} \cup \mathbb{T}^d\}^N} |(\mathcal{E}_1 \circ \mathcal{G} \circ \mathcal{E}_1^+)^{(k)} - \mathcal{E}_1 \circ \mathcal{G}^{(k)} \circ \mathcal{E}_1^+|^2 d\mathcal{E}_{1*}\mu \leq C \left( \sum_{p > N/C\log(N)} \lambda_p \right)$$

Finally, for the last term, similar to the above,

$$\int_{\mathbb{R} \cup \mathbb{T}^{dN}} |\mathcal{D}_2^+ \circ \mathcal{E}_1^+ \circ \mathcal{E}_1 \circ \mathcal{G}^{(k)} \circ \mathcal{E}_1^+ - \mathcal{D}_2 \circ \mathcal{G}^{(k)} \circ \mathcal{E}_1^+|^2 dy$$

$$\leq \int_{H^s(\mathbb{T}^d)} |\mathcal{D}_2^+ \circ \mathcal{E}_1^+ \circ \mathcal{E}_1 - \mathcal{D}_2^+|^2 d\mathcal{G}_*^{(k)}\mu + C_{\mathcal{D}_2^+}^2 C_{\mathcal{G}}^{2k}(C_{\mathcal{E}_1^+}C_{\mathcal{E}_1} + 1)^2 \int_{H^s(\mathbb{T}^d)} \|\mathcal{E}_1^+ \circ \mathcal{E}_1 - id\|^2 d\mu$$

$$\leq \int_{H^s(\mathbb{T}^d)} |\mathcal{E}_1^+ \circ \mathcal{E}_1 - id|^2 d\mathcal{G}_*^{(k)}\mu + C_{\mathcal{D}_2^+}^2 C_{\mathcal{G}}^{2k}(C_{\mathcal{E}_1^+}C_{\mathcal{E}_1} + 1)^2 \int_{H^s(\mathbb{T}^d)} \|\mathcal{E}_1^+ \circ \mathcal{E}_1 - id\|^2 d\mu.$$

As in the estimates of the third term, we can get analogous results. When we put the results together, we can derive the desired theorem. □

## B.2 Proof of Theorem 2

We construct the $\mathcal{G}_{\text{graph}}$ by using the fully connected graph neural network in Definition 2. When discussing the solution operator in a periodic domain, we concentrate on the graph and the distances between points within this domain. Specifically, we consider node features $\{(u^0(\boldsymbol{x}_i), \boldsymbol{x}_i)\}_{i=0}^{N-1}$ and the corresponding distance $w_{ij} = \|\boldsymbol{x}_i - \boldsymbol{x}_j\|_{l^2(\mathbb{T}^d)}$. This distance is calculated as the minimum of the set $\{\|(x_{i_{(1)}}, \ldots, x_{i_{(d)}}) - (x_{j_{(1)}}, \ldots, x_{j_{(d)}})\|_{l^2(\mathbb{T}^d)}, \ldots, \|(1 - x_{i_{(1)}}, \ldots, 1 - x_{i_{(d)}}) - (1 - x_{j_{(1)}}, \ldots, 1 - x_{j_{(d)}})\|_{l^2(\mathbb{T}^d)}\}.$.

$\mathcal{G}_{\text{graph}}$ generates a sequence of functions $\{(h_0^{(k)}, \ldots, h_{N-1}^{(k)})\}_{k=0}^{\infty}$ by the given manner. For a finite fixed $M$, we approximate the value of $\{u^1(\boldsymbol{x}_i)\}_{i=0}^{N-1}$ with $h_i^{(M)}(\{(u^0(\boldsymbol{x}_i), \boldsymbol{x}_i)\}_{i=0}^{N-1})$. The error for given $\mathcal{G}_{\text{graph}}$ with respect to the probability measure $\mu$ can be defined by

$$\left\| \mathcal{G}^{(1)}(u_0) - \mathcal{G}_{\text{graph}}(u_0) \right\|_{L^2(\mu)} := \int_{H^s(\mathbb{T}^d)} \sum_{i=0}^{N-1} (u^1(\boldsymbol{x}_i) - h_i^{(M)}(\{(u^0(\boldsymbol{x}_i), \boldsymbol{x}_i)\}_{i=0}^{N-1}))^2 d\mu.$$

*Proof.* Consider the periodic domain $\mathbb{T}^d$, which includes points $\boldsymbol{x} = (x_{(1)}, x_{(2)}, \ldots, x_{(d)})$ within $\mathbb{T}^d$. The initial function is given by $f(\boldsymbol{x}) = f(x_{(1)}, x_{(2)}, \ldots, x_{(d)})$. We define the rotation map $R_t : \mathbb{T}^d \to \mathbb{T}^d$ on the coordinates as:

$$R_t \begin{pmatrix} x_{(1)} \\ x_{(2)} \\ x_{(3)} \\ \vdots \\ x_{(d)} \end{pmatrix} = \begin{pmatrix} \cos(2\pi t) & -\sin(2\pi t) & 0 & \cdots & 0 \\ \sin(2\pi t) & \cos(2\pi t) & 0 & \cdots & 0 \\ 0 & 0 & 1 & \cdots & 0 \\ \vdots & \vdots & \vdots & \ddots & \vdots \\ 0 & 0 & 0 & \cdots & 1 \end{pmatrix} \begin{pmatrix} x_{(1)} - \frac{1}{2} \\ x_{(2)} - \frac{1}{2} \\ x_{(3)} \\ \vdots \\ x_{(d)} \end{pmatrix} + \begin{pmatrix} \frac{1}{2} \\ \frac{1}{2} \\ 0 \\ \vdots \\ 0 \end{pmatrix}.$$

Consequently, the resulting rotated function $g(t, \boldsymbol{x}) = f(R_t(\boldsymbol{x}))$ represents the state of $f$ after a rotation by an angle $t$ around the axis through $(1/2, 1/2, 0, \ldots, 0)$ in the first two coordinate planes.

We then establish the mapping $\mathcal{G} : H^s(\mathbb{T}^d) \to H^s(\mathbb{T}^d)$ as follows:

$$\mathcal{G}(f(\boldsymbol{x})) := f(R_{1/2}(\boldsymbol{x})), \tag{12}$$

for any function $f$ in the space $H^s(\mathbb{T}^d)$.

Define the set $A$ by

$$A := \left( \left\{ \left[ \frac{1}{12}, \frac{3}{12} \right] \bigcup \left[ \frac{5}{12}, \frac{7}{12} \right] \bigcup \left[ \frac{9}{12}, \frac{11}{12} \right] \right\} \right)^d. \tag{13}$$

Using the smooth version of Urysohn's lemma, we're able to construct the following two functions.

1. A smooth function $f_1$ that equals 1 on the $d$-dimensional interval $\left[ \frac{5}{12}, \frac{7}{12} \right]^d$ and 0 on the other rectangular sections within set $A$.

2. Another smooth function $f_2$ that is 1 on the $d$-dimensional interval $\left[ \frac{1}{12}, \frac{3}{12} \right]^d$ and 0 elsewhere within $A$.

Let's consider a mesh within the set $A$ that follows a specific property related to the translation of grid points.

- For any grid point $\boldsymbol{x}_i = (x_{i_{(1)}}, \ldots, x_{i_{(d)}})$, and for any arbitrary integers $n_1, \ldots, n_d$, the point $(x_{i_{(1)}} + n_1/3 \pmod 1, \quad \ldots, x_{i_{(d)}} + n_d/3 \pmod 1)$ is also selected as a grid point.

It is noteworthy that we can choose such grid points with a nonzero probability. Now We consider the measure $\mu$ on the set of initial conditions where We would like to select $f_1$ or $f_2$ by the input function $f$ with probability 1/2 respectively.

Let's consider using graph neural networks, denoted as $\mathcal{G}_{graph}$ following the definition in 2. As for the message passing algorithm, the input for each node does not change whether we apply the function $f_1$ to grid points within $\left[ \frac{5}{12}, \frac{7}{12} \right]^d$ or $f_2$ to those within $\left[ \frac{1}{12}, \frac{3}{12} \right]^d$. Consequently, the graph neural network's predictions should remain consistent.

On the one hand, $\mathcal{G}$ is constructed to map a value of 1 to 1 when using $f_1$ as input function and from 1 to 0 when using $f_2$, for each corresponding grid point. Therefore,

$$\|\mathcal{G}(u_0) - \mathcal{G}_{\text{graph}}(u_0)\|_{L^2(\mu)} \geq (1/2(\text{output} - 0)^2 + 1/2(\text{output} - 1)^2)^{1/2} = 1/2.$$

$\square$

The readers might wonder why the statement of neural networks related to our model and graph neural networks are different. Note that there exist only two eigenfunctions and all eigenvalues $\lambda_j$ except finite many should be zero by construction.

Therefore, the term $C(\sum_{j>N/C(\log(N))} \lambda_j)^{1/2}$ in the right-hand side should be zero for large $N$. And it is obvious that $CN^{-s/d}$ goes to 0 when $N$ is large.

In conclusion, we can say that there exists a Lipschitz continuous operator $\mathcal{G}^{(1)} : H^s(\mathbb{T}^d) \to H^s(\mathbb{T}^d)$ such that for any $C \in \mathbb{N}$, there exist infinitely many $N \in \mathbb{N}$ satisfying the following inequality with a non-zero probability $\delta > 0$:

$$\left\| \mathcal{G}^{(1)}(u_0) - \mathcal{G}_{\text{graph}}(u_0) \right\|_{L^2(\mu)} \geq C \left( \sum_{j>N/C\log(N)} \lambda_j \right)^{1/2} + CN^{-s/d}.$$

## C  DETAILS ON EXPERIMENTS AND ADDITIONAL EXPERIMENTS

### C.1  DETAIL SETTING ON GRAPH

The edges $(i, j) \in \mathcal{E}$ are constructed based on the proximity of node positions, connecting nodes within a specified distance. In actual experiments, we considered nodes as grids with given initial conditions. There are broadly two methods for defining edges. One approach involves setting a threshold based on the distances between grids in the domain, connecting edges if the distance between these grids is either greater or smaller than the specified threshold value. Another method involves utilizing classification techniques, such as the $k$-nearest neighbors ($k$-NN) algorithm, to determine whether to establish an edge connection. We determined whether to connect edges based on the $k$-NN algorithm with $k =6$ for 1D, $k = 8$ for 2D. Therefore, the processing of $\phi$ and $\psi$ takes place based on these edges. The crucial point here is that once the Graph $G = (\mathcal{V}, \mathcal{E})$ is constructed according to a predetermined criterion, even with a different set of sensor points, $\phi$ and $\psi$ remain unchanged as processor networks applied to the respective nodes and their connecting edges.

### C.2  DATASET

Similar to other graph-based PDE solver studies (Brandstetter et al., 2022; Boussif et al., 2022), we consider the 1D Burgers' equation as

$$\partial_t u + \partial_x(\alpha u^2 - \beta \partial_x u + \gamma \partial_{xx} u) = \delta(t, x), \quad t \in T = [0, 4], x \in \Omega = [0, 16),$$
$$u(0, x) = \delta(0, x), \quad x \in \Omega, \tag{14}$$

where $\delta(t, x)$ is randomly generated as

$$\delta(t, x) = \sum_{j=1}^{5} A_j \sin(a_j t + b_j x + \phi_j) \tag{15}$$

where $a_j$, $b_j$ and $\phi_j$ are uniformly sampled as

$$A_j \in \left[-\frac{1}{2}, \frac{1}{2}\right], a_j \in \left[-\frac{2}{5}, \frac{2}{5}\right], b_j \in \left\{\frac{\pi}{8}, \frac{2\pi}{8}, \frac{3\pi}{8}\right\}, \phi_j \in [0, 2\pi]. \tag{16}$$

We conducted a direct comparison with the models using the data E1, E2, and E3 as provided in Brandstetter et al. (2022); Boussif et al. (2022). For a more detailed understanding of the data, refer to those studies.

Also, we take the 2D shallow water equation data from Takamoto et al. (2022). The shallow water equations, which stem from the general Navier-Stokes equations, provide a suitable framework for the modeling of free-surface flow problems. In two dimensions, it can be expressed as the following system of hyperbolic PDEs

$$\frac{\partial h}{\partial t} + \frac{\partial}{\partial x}(hu) + \frac{\partial}{\partial y}(hv) = 0,$$

$$\frac{\partial(hu)}{\partial t} + \frac{\partial}{\partial x}\left(u^2 h + \frac{1}{2}gh^2\right) + \frac{\partial}{\partial y}(huv) = 0, \quad t \in [0, 1], \boldsymbol{x} = (x, y) \in \Omega = [-2.5, 2.5]^2$$

$$\frac{\partial(hv)}{\partial t} + \frac{\partial}{\partial y}\left(v^2 h + \frac{1}{2}gh^2\right) + \frac{\partial}{\partial x}(huv) = 0,$$

$$h(0, x, y) = h_0(x, y), \tag{17}$$

where $h(t, x, y)$ is the height of water with horizontal and vertical velocity $(u, v)$ and $g$ is the gravitational acceleration. We generate the random samples of initial conditions similar to the setting of Takamoto et al. (2022). The initial condition is generated by

$$h_0(x, y) = \begin{cases} 2.0, & \text{for } r < \sqrt{x^2 + y^2} \\ 1.0, & \text{for } r \geq \sqrt{x^2 + y^2} \end{cases} \tag{18}$$

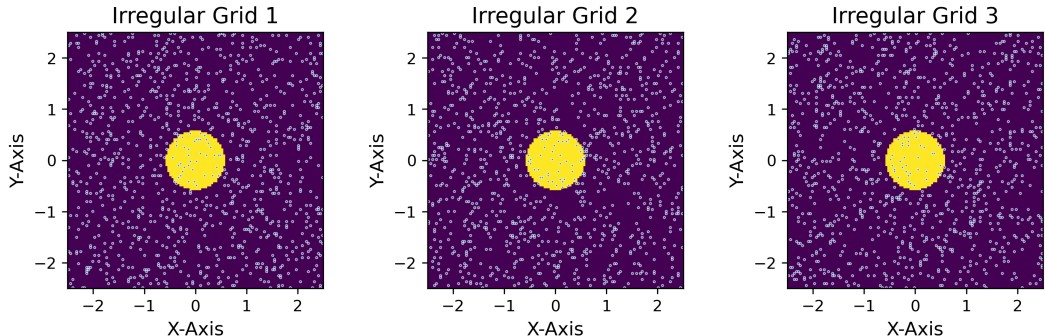

Figure 6: Three types of irregular grid (Irregular I, Irregular II, and Irregular III) used to train the models in shallow water eqaution

Table 4: Mean Rel. $L^2$ test errors for N-S equation data using regular/irregular sensor points using GDON(ours) and FFNO.

| Data | Type of Sensor Points | GDON(ours) | FFNO |
|---|---|---|---|
| 2D N-S | Regular | 0.1287 | 0.5945 |
| | Irregular I | 0.1243 | 0.5841 |
| | Irregular II | 0.1257 | 0.5907 |
| | Irregular III | 0.1289 | 0.5972 |

where the radius $r$ is uniformly sampled from $[0.3, 0.7]$.

We utilize the same Navier-Stokes data employed in Li et al. (2020). The dynamics of a viscous fluid are described by the Navier-Stokes equation. In the vorticity formulation, the incompressible Navier-Stokes equation on the unit torus can be represented as follows:

$$\begin{cases} \frac{\partial w}{\partial t} + u \cdot \nabla w - \nu \Delta w = f, & (t, \boldsymbol{x}) \in [0, T] \times (0, 1)^2, \\ \nabla \cdot u = 0, & (t, \boldsymbol{x}) \in [0, T] \times (0, 1)^2, \\ w(0, \boldsymbol{x}) = w_0(\boldsymbol{x}), & \boldsymbol{x} \in (0, 1)^2, \end{cases} \tag{19}$$

Here, $w$, $u$, $\nu$, and $f$ represent the vorticity, velocity field, viscosity, and external force, respectively.

### C.3    COMPARISON WITH F-FNO FOR THE N-S EQUATION

Our focus is on comparing our model with existing graph neural network(GNN)-based models capable of simulating time-dependent PDEs on irregular domains, such as MP-PDE and MAgNet. Consequently, instead of considering variations of FNO, we concentrated on GNN-based PDE solvers for experiment baseline. Therefore, FNO, being a fundamental model in operator learning area, was compared only on regular grids. We included experiments comparing our model with F-FNO proposed in Tran et al. (2023), which is state-of-the-art on regular grids and applicable to irregular grids. As shown in Table 4, the F-FNO is applicable to irregular grids of N-S equation data, but it generally exhibits higher errors compared to GraphDeepONet. This is attributed to the limited capacity for the number of input features. We used the F-FNO model, which is built for Point Cloud data, to predict how solutions will evolve over time. At first, the model was designed to process dozens of input features, which made it difficult to include all the initial values from a two-dimensional grid.

Table 5: The training time and inference time for the N-S equation data using GNN based models.

| Data | Model | Training time per epoch (s) | Inference time per timestep (ms) |
|---|---|---|---|
| Navier Stokes (2D Irregular I) | GDON(Ours) | 7.757 | 19.49 |
| | MAgNet | 18.09 | 48.81 |
| | MP-PDE | 4.88 | 10.1 |

## C.4 COMPUTATIONAL TIME COMPARISON WITH BENCHMARK MODELS

One significant advantage of models based on Graph Neural Networks (GNNs), such as MP-PDE, MAgNet, and GraphDeepONet (ours), compared to traditional numerical methods for solving time-dependent PDEs, lies in their efficiency during inference. In traditional numerical methods, solving PDEs for different initial conditions requires recalculating the entire PDE, and in real-time weather prediction scenarios (Kurth et al., 2022), where numerous PDEs with different initial conditions must be solved simultaneously, this can result in a substantial computational burden. On the other hand, models based on GNNs (MP-PDE, MAgNet, GraphDeepONet), including the process of learning the operator, require data for a few frames of PDE. However, after training, they enable rapid inference, allowing real-time PDE solving. More details on advantage using operator learning model compared to traiditional numerical method is explained in many studies (Goswami et al., 2022; Kovachki et al., 2021b).

Table 5 presents a computational time comparison between our proposed GraphDeepONet and other GNN-based models. Due to its incorporation of global interaction using (8) for a better understanding of irregular grids, GraphDeepONet takes longer during both training and inference compared to MP-PDE. However, the MAgNet model, which requires separate interpolation for irregular grids, takes even more time than MP-PDE and GraphDeepONet. This illustrates that our GraphDeepONet model exhibits a trade-off, demonstrating a stable accuracy for irregular grids compared to MP-PDE, while requiring less time than MAgNet.

## C.5 MODEL HYPERPARAMETERS FOR BENCHMARK MODELS AND OUR MODEL

We trained various models, including DeepONet and VIDON, following the architecture and sizes as well as the training hyperparameters outlined in Prasthofer et al. (2022). Additionally, MP-PDE and MAgNet utilized parameter settings as provided in Boussif et al. (2022) without modification. We trained our model, the GraphDeepONet, using the Adam optimizer, starting with an initial learning rate of 0.0005. This learning rate is reduced by 20

In the small architecture, the encoder was set up with a width of 128 and a depth of 2 for epsilon. The processor components, $\phi$, and $\psi$, each had a width of 128 and a depth of 2. We employed distinct $\phi$ and $\psi$ for each of the three message-passing steps. In the decoder, we assigned $\omega_{\text{gate}}$ and $\omega_{\text{feature}}$ for aggregation to the neural network, which had a width of 128 and a depth of 3. The trunk net, $\boldsymbol{\tau}$, was configured with a width of 128 and a depth of 3.

For the large architecture, the width of all neural networks was set to 128, and the depth was set to 3, except for the trunk net. The trunk net's depth was set to 5. The number of message-passing steps was set to 3. For more specific details, refer to the code.

## C.6 ENFORCING BOUNDARY CONDITION USING THE GRAPHDEEPONET

Utilizing the structure of DeepONet enables us to enforce the boundary condition $B[u] = 0$ as hard constraints. To elaborate further, we impose hard constraints for periodic boundary conditions and Dirichlet through a modified trunk net, which is one of the significant advantages of the DeepONet model structure, as also explained in [1]. For instance, in our paper, we specifically address enforcing periodic boundary conditions in the domain $\Omega$. To achieve this, we replace the network input $x$ in the trunk net with Fourier basis functions $\left(1, \cos(\frac{2\pi}{|\Omega|}x), \sin(\frac{2\pi}{|\Omega|}x), \cos(2\frac{2\pi}{|\Omega|}x), ...\right)$, naturally leading to a solution $u(t, \boldsymbol{x})$ $(\boldsymbol{x} \in \Omega)$ that satisfies the $|\Omega|$-periodicity. As depicted in Figure 7, the results reveal that while other models fail to perfectly match the periodic boundary conditions, GraphDeepONet successfully aligns with the boundary conditions.

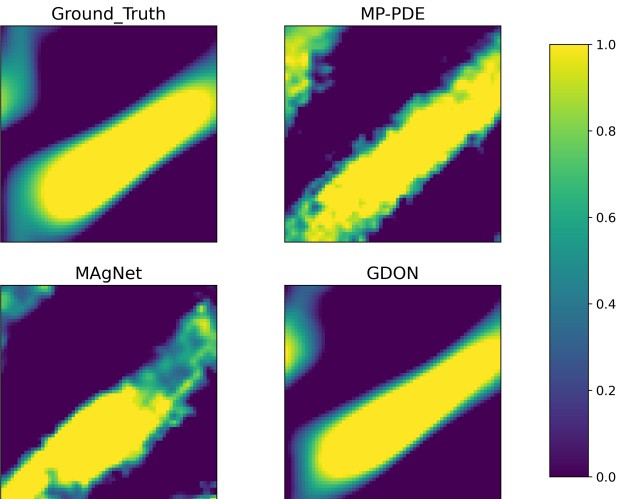

Figure 7: One snapshot for N-S equation data using MP-PDE, MAgNet, and GraphDeepONet.

While our experiment primarily focuses on periodic boundary conditions, it is feasible to handle Dirichlet boundaries as well using ansatz extension as discussed in Choudhary et al. (2020); Horie & Mitsume (2022). If we aim to enforce the solution $\widetilde{u}(t, \boldsymbol{x}) = g(\boldsymbol{x})$ at $\boldsymbol{x} \in \partial\Omega$, we can construct the following solution:

$$\widetilde{u}(t, \boldsymbol{x}) = g(\boldsymbol{x}) + l(\boldsymbol{x}) \sum_{j=1}^{p} \nu_j [\boldsymbol{f}_{0:N-1}^1, \Delta t] \tau_j(\boldsymbol{x})$$

where $l(\boldsymbol{x})$ satisfies

$$\begin{cases} l(\boldsymbol{x}) = 0, & \boldsymbol{x} \in \partial\Omega, \\ l(\boldsymbol{x}) > 0, & \text{others.} \end{cases}$$

By constructing $g(\boldsymbol{x})$ and $l(\boldsymbol{x})$ appropriately, as described, we can effectively enforce Dirichlet boundary conditions as well. While the expressivity of the solution using neural networks may be somewhat reduced, there is a trade-off between enforcing boundaries and expressivity.

### C.7 EXPERIMENTS ON BURGERS' EQUATION

For Burgers' equation, we generate the uniform grid of 50 points in $[0, 16]$. We divided the time interval from 0 to 4 seconds uniformly to create 250 time steps. We started with 25 initial values for each segment, then predicted the values for the next 25 instances, and so on. The total number of prediction steps is 9, calculated by dividing 225 by 25. In all experiments, we used a batch size of 16.

The training data consisted of 1896 samples, while both the validation and test samples contained 128 samples each. For irregular data, we selected 50 points from a uniform distribution over 100 uniform points within the range of 0 to 16 and made predictions on a fixed grid. The number of samples is the same as in the regular data scenario. To ensure a fair comparison in time extrapolation experiments, each model was assigned to learn the relative test error with a precision of 0.2 concerning the validation data. Our model conclusively shows superior extrapolation abilities compared to VIDON and DeepONet. Unlike DeepONet and VIDON, which tended to yield similar values throughout all locations after a given period, our model effectively predicted the local propagation of values.

In Table 1, our model consistently outperforms other graph neural networks, regardless of the dataset or grid regularity. Figure 8 illustrates the solution profiles and prediction derived from the models. It's important to highlight that the shape of the solution progressively sharpens due to the nature of Burgers' equation. Our model can make more accurate predictions for approximating the sharp solution compared to other models. Notably, MAgNet can generate the smoothest prediction, learning the global representation through a neural network.

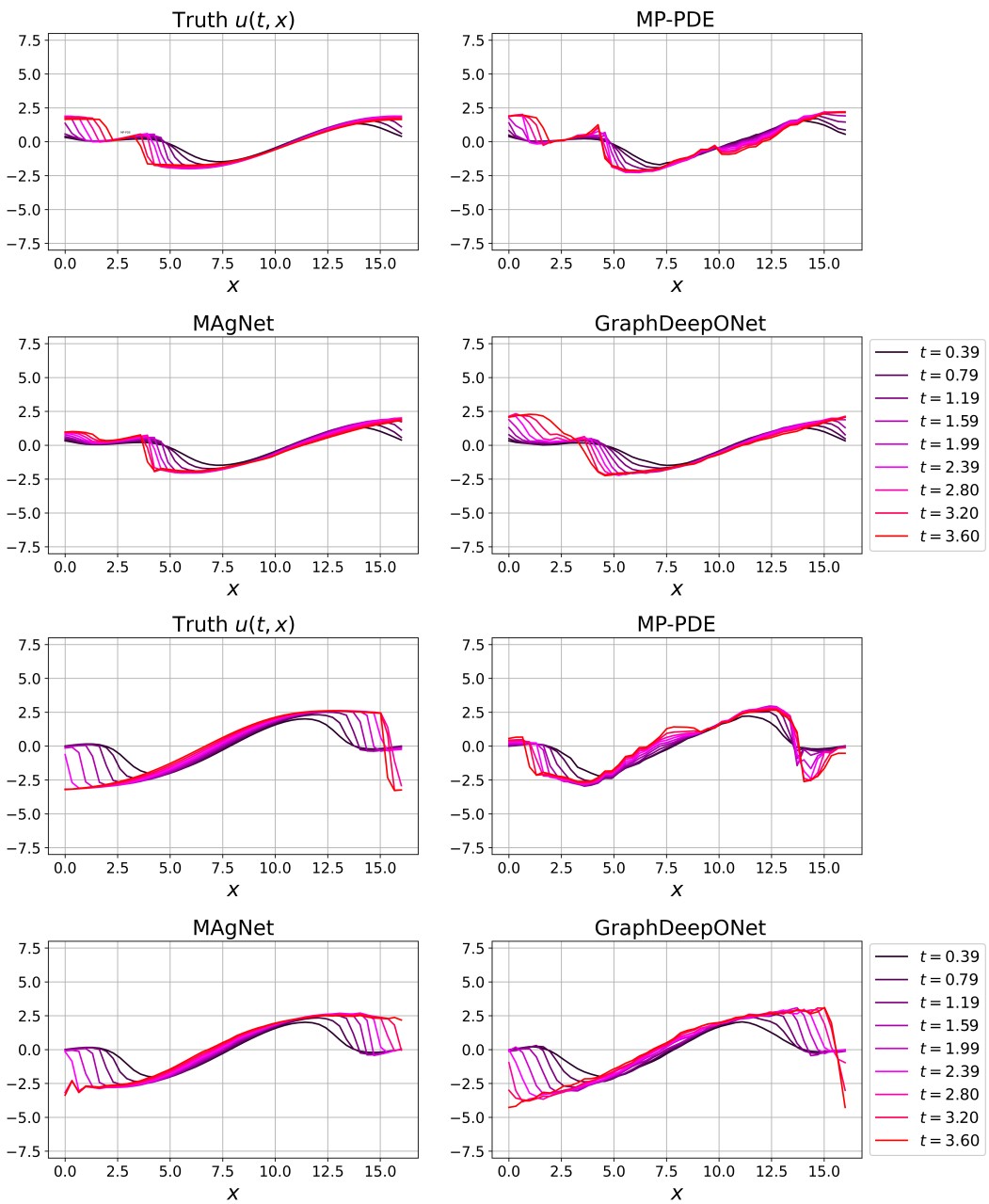

Figure 8: Ground truth solution and two prediction profiles for Burgers' equation on a uniform grid.

In Figure 3, GraphDeepONet was trained with a parameter setting of $K = 25$ for Burgers data where it divided time $t$ into 250 intervals within the range $t \in [0, 4]$. The model received inputs accordingly and predicted the solutions for the next 25 frames based on the solutions of the preceding 25 frames. While this grouping strategy led to good accuracy, the results in Figure 3 show a noticeable discontinuity every 25 frames as your concern. However, as illustrated in Figure 9, grouping frames into smaller units with $K = 5$ results in a smoother prediction appearance although the error is slightly increased compared to the $K = 25$ case. This trade-off in GraphDeepONet is an essential aspect of our method.

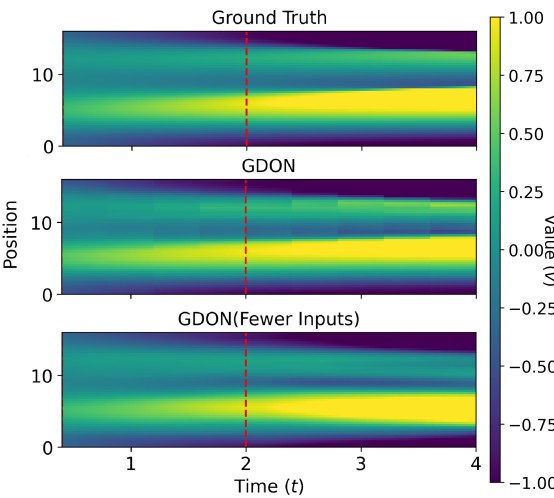

Figure 9: Comparison of solution profiles obtained from the Burgers' equation time extrapolation simulations using GraphDeepONet, with $K = 25$ and $K = 5$.

### C.8   EXPERIMENTS ON 2D SHALLOW WATER EQUATION AND 2D N-S EQUATION

For the 2D shallow water equation, we generate the grid of $1024 = 32^2$ points for the regular setting. For irregular data, we selected an equal number of points from a uniform distribution over $128^2$ points within the rectangle $[-2.5, 2.5]^2$ and made predictions on a fixed grid. Figure 6 illustrates how we set up irregular sensor points for training GNN-based models and our model.

We evenly divided the time interval from 0 to 1 second uniformly to create 101 time steps. We started with 10 initial values for each segment, then predicted the values for the next 10 instances, and so on. The total number of prediction steps is 9, calculated by dividing 101-1=100 by 10. We remark that the values at $t = 1$ were excluded from the data set. In all experiments, we used a batch size of 4. For both regular data and irregular data, the training data consisted of 600 samples, while both the validation and test samples contained 200 samples each. Note that the MAgNet has the capability to interpolate values using the neural implicit neural representation technique. However, we did not utilize this technique when generating Figure 4, which assesses the interpolation ability for irregular data. For clarity, we've provided Figure 10 the predictions on the original irregular grid prior to interpolation.

In reference to the 2D Navier-Stokes equation, we apply the data from Li et al. (2020) with a viscosity of 0.001. For regular data, we generate the grid of $1024 = 32^2$ points. For irregular data, we selected an equal number of points from a uniform distribution over $64^2$ points within the rectangle $[0, 1]^2$ and made predictions on a fixed grid. We evenly divided the time interval from 1 to 50 seconds uniformly to create 50 time steps. We started with 10 initial values for each segment, then predicted the values for the next 10 instances, and so on. The total number of prediction steps is 4, calculated by dividing 50-10=40 by 10. In all experiments, we used a batch size of 4. For both regular data and irregular data, the training data consisted of 600 samples, while both the validation and test samples contained 200 samples each.

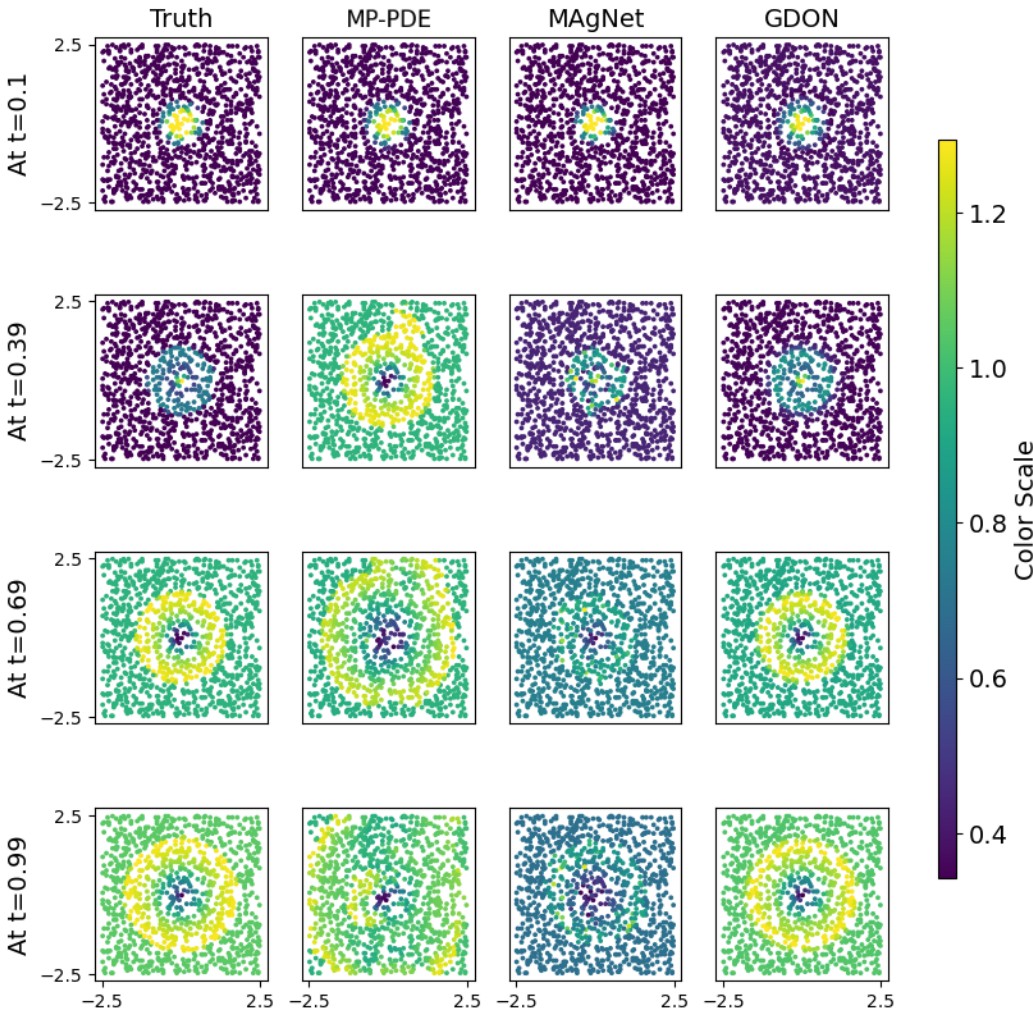

Figure 10: Ground truth solution and prediction profile for the 2D shallow water equation on a irregular grid.

