# OpenReview forum: "GraphDeepONet: Learning to simulate time-dependent partial differential equations using graph neural network and deep operator network"
_ICLR.cc/2024/Conference — Submitted to ICLR 2024_

### Official Review · Reviewer_9eTr · 2023-10-30

**Soundness:** 2 fair
**Presentation:** 3 good
**Contribution:** 3 good
**Rating:** 6
**Confidence:** 3

**Summary:**

This paper proposes GraphDeepONet, an autoregressive model that combine DeepONet and GNN to solve partial differential equation. GraphDeepONet outperforms existing GNN-based neural PDE solvers on both regular and irregular grids. By inheriting the advantages from DeepONet, the proposed method can perform prediction on arbitrary grids. Moreover, this method is able to do time extrapolation which is not feasible in the traditional DeepONet.

**Strengths:**

- This method incorporates time information into the branch net using a GNN and enables time extrapolation prediction for PDE.
- Compared to other GNN-based PDE solvers, GraphDeepONet shows superior accuracy in predicting solution at arbitrary positions on irregular grids.
- The proposed method has a theoretical guarantee that it is universally capable of approximating continuous operators for arbitrary time intervals.
- Overall this paper is well-written, and the presentation is clear and easy to follow.

**Weaknesses:**

The evaluation of the proposed methods is not enough, specifically:
- 1D Burger equation and 2D shallow water are much easier PDEs. In most papers, 2D incompressible Navier-Stokes equation is considered as a test PDE. I highly recommend authors to add experiments on 2D NS equation to show the effectiveness of their method.
- In table 1, the performance of GraphDeepONet are consistently worse than FNO-2D, and FNO-2D is not a recent model on regular grid. This could be a major weakness. I recommend authors compare their method with state-of-the-art model on regular grids, such as FFNO [2] and GFNO [3].
- For irregular domain, there are several FNO-based method can handle irregular grids, such as GeoFNO [1] and FFNO[2]. For completeness, I think these methods should also be considered as baselines on irregular grids.

[1] Li, Zongyi, et al. "Fourier neural operator with learned deformations for pdes on general geometries." arXiv preprint arXiv:2207.05209 (2022).\
[2] Tran, Alasdair, et al. "Factorized fourier neural operators." arXiv preprint arXiv:2111.13802 (2021). \
[3] Helwig, Jacob, et al. "Group Equivariant Fourier Neural Operators for Partial Differential Equations." arXiv preprint arXiv:2306.05697 (2023).

**Questions:**

see weakness part

---

> ### Author Response · Authors · 2023-11-23
> **Response to Reviewer 9eTr**
>
> We appreciate your constructive comments and suggestions. Your comments gave us a good opportunity to review and verify the contents of the paper. We summarize your comments and respond to your concerns below.
>
> > **Q1.** I highly recommend authors to add experiments on 2D NS equation to show the effectiveness of their method.
>
> **A.** Thank you for the suggestion. As you suggested, it would be beneficial to enhance the demonstration of the strengths of our GraphDeepONet by incorporating experiments on more challenging datasets commonly used in the field of operator learning. We added experiments on the more complex dataset: 2D Navier-Stokes equations. We utilized data from the setting of the 2D Navier-Stokes equations for a viscous, incompressible fluid in vorticity form on the unit torus, as employed in [1,2]. As can be seen in Table 2, our approach exhibits similar trends for the N-S equation across different datasets. While other methods show variations in errors depending on the type of irregular grid, GraphDeepONet demonstrates good stability across irregular grids.
>
> > **Q2.** I recommend authors compare their method with sota model on regular grids, such as F-FNO [2] and G-FNO [3], and with model on irregular grids, such as geo-FNO [1] and F-FNO[2].
>
> **A.** Thank you for suggesting relevant baselines for comparison. Our focus is on comparing our model with existing graph neural network(GNN)-based models capable of simulating time-dependent PDEs on irregular domains, such as MP-PDE and MAgNet. Consequently, instead of considering variations of FNO, we concentrated on GNN-based PDE solvers for experiment baseline. Therefore, FNO, being a fundamental model in operator learning area, was compared only on regular grids. However, as your insightful suggestion, we have added comparisons with modified FNO models applicable to both regular and irregular grids, providing significant enhancement to the paper. Specifically, we included experiments comparing our model with F-FNO proposed in [2], which is state-of-the-art on regular grids and applicable to irregular grids. As shown in Table 4, the F-FNO is applicable to irregular grids of N-S equation data, but it generally exhibits higher errors compared to GraphDeepONet. This is attributed to the limited capacity for the number of input features. We used the F-FNO model, which is built for Point Cloud data, to predict how solutions will evolve over time. At first, the model was designed to process dozens of input features, which made it difficult to include all the initial values from a two-dimensional grid. Furthermore, we have added references to geo-FNO [1] and G-FNO [3], as suggested by you, to highlight their relevance to our model in the main text. We added this explanation in Appendix C.3.
>
> *[1] Li, Zongyi, et al. "Fourier neural operator with learned deformations for pdes on general geometries." arXiv preprint arXiv:2207.05209 (2022).*
>
> *[2] Tran, Alasdair, et al. "Factorized fourier neural operators." arXiv preprint arXiv:2111.13802 (2021).*
>
> *[3] Helwig, Jacob, et al. "Group Equivariant Fourier Neural Operators for Partial Differential Equations." arXiv preprint arXiv:2306.05697 (2023).*

---

### Official Review · Reviewer_XsTS · 2023-10-30

**Soundness:** 3 good
**Presentation:** 3 good
**Contribution:** 3 good
**Rating:** 6
**Confidence:** 3

**Summary:**

The paper proposes GraphDeepONet (GDON), which extends the capacity of DeepONet [Lu+ 2019] to enable extrapolating time evolution. The authors suggest incorporating temporal information into the branch net instead of the trunk net. The mathematical analysis shows that the method still maintains the universal approximation property. In addition, theorem 2 shows that graph-based models can fail to approximate a solution to the considered problem, implying the superiority of the proposed operator-based approach that enables interpolation in space. The experimental results demonstrate that the proposed approach has good stability regarding irregular grids.

**Strengths:**

* The method extends operator learning approaches by separating dependencies of time and space into branch net and trunk net, respectively.
* The theoretical analysis supports the methods' expressibility and superiority compared to graph-based models.
* Experimental results show that the method has good stability regarding irregular grids, which is preferable in a realistic application.

**Weaknesses:**

* The experiments seem not to be complete. The method incorporates global interaction using Equation 8. Therefore, it would be fair to compare with graph neural networks with varying numbers of hops visible to GNNs. In addition, the authors should show the computation time-accuracy tradeoff because the method seems to have a massive computation cost.
* Related to the first point, it is not clear to use machine learning over classical numerical analyses. The authors should clarify the superiority of the method compared to classical analysis, e.g., in terms of computational time.
* The paper states that the method can predict the solution for any x in R^d (Section 3.5). However, the reviewer believes the method can interpolate in space but extrapolate. Therefore, the reviewer suggests the authors weaken the statement, e.g., x in \Omega. Otherwise, the authors should demonstrate that the method can extrapolate in space.

### Minor points
* In Section 1, the term "Trunk net" is used without explanation, which could be unclear for readers unfamiliar with DeepONet. Therefore, the authors could add some explanations about it or not touch such details in the introduction.
* In Table 1, \pi could be \pm.
* Section 4 (Comparison with GNN-based PDE-solvers): "2 summarizes" -> "Table 2 summarizes"

**Questions:**

* The paper states that the method can deal with periodic boundary conditions. How about other ones (e.g., Dirichlet and Neumann)?
* By looking at Figure 3, the proposed method gives a non-smooth solution in time and space. Why does it happen and how to improve the situation? The smoothness of the solution is sometimes essential for downstream tasks if we would like to compute the derivative of the solution.
* By MPNN, the authors mean MP-PDE [Brandstetter+ 2022]? In my understanding, MPNN usually refers to the message-passing neural networks [Glimer+ 2017]. If the authors mean [Brandstetter+ 2022], the reviewer recommends using MP-PDE or other appropriate abbreviations different from MPNN. If the authors did not compare with MP-PDE, the reviewer strongly recommends doing it because the dataset is extracted from that work.

---

> ### Author Response · Authors · 2023-11-23
> **Response to Reviewer XsTS - 1**
>
> We appreciate your positive comments. Your comments gave us a good opportunity to review and verify the contents of the paper. We summarize your comments and respond to your concerns below.
>
> > **Q1.** The method incorporates global interaction using Equation 8. The authors should show the computation time-accuracy tradeoff because the method seems to have a massive computation cost. The authors should clarify the superiority of the method compared to classical analysis, e.g., in terms of computational time.
>
> **A.** Thank you for your valuable feedback. One significant advantage of models based on Graph Neural Networks (GNNs), such as MP-PDE, MAgNet, and GraphDeepONet (ours), compared to traditional numerical methods for solving time-dependent PDEs, lies in their efficiency during inference. In traditional numerical methods, solving PDEs for different initial conditions requires recalculating the entire PDE, and in real-time weather prediction scenarios [1], where numerous PDEs with different initial conditions must be solved simultaneously, this can result in a substantial computational burden. On the other hand, models based on GNNs (MP-PDE, MAgNet, GraphDeepONet), including the process of learning the operator, require data for a few frames of PDE. However, after training, they enable rapid inference, allowing real-time PDE solving. More details on advantage using operator learning model compared to traiditional numerical method is explained in many studies [2,3].
>
> Furthermore, it is very good point to compare computational time of our model with the other GNN-based models. Table 5 presents a computational time comparison between our proposed GraphDeepONet and other GNN-based models. Due to its incorporation of global interaction using Equation 8 for a better understanding of irregular grids, GraphDeepONet takes longer during both training and inference compared to MP-PDE. However, the MAgNet model, which requires separate interpolation for irregular grids, takes even more time than MP-PDE and GraphDeepONet. This illustrates that our GraphDeepONet model exhibits a trade-off, demonstrating a stable accuracy for irregular grids compared to MP-PDE, while requiring less time than MAgNet.
>
> > **Q2.** Therefore, the reviewer suggests the authors weaken the statement, e.g., $\boldsymbol{x}\in\Omega$.
>
> **A.** Thank you for pointing out our mistake. As you correctly noted, our method cannot predict the solution for any $\boldsymbol{x}$ in whole $\mathbb{R}^d$. It is more accurate to state that our approach, as you rightly understood, allows for extrapolation in time $t$ and for interpolation in the spatial domain $\boldsymbol{x}$ at arbitrary positions within the learning domain $\boldsymbol{x}\in\Omega\subset\mathbb{R}^d$ when predicting the solution $u(t,\boldsymbol{x})$. In response to this, we have used $\Omega$ in the paper to more accurately represent the spatial domain and have adjusted the related explanations to avoid confusion. Thanks for your insightful feedback.
>
> > **Q3.** The term "Trunk net" is used without explanation, which could be unclear for readers unfamiliar with DeepONet.
>
> **A.** Thank you for pointing out the missing explanation. As you mentioned, it was indeed a lapse in clarity to introduce the trunk net in the introduction without providing an explanation. This could make it challenging for readers unfamiliar with DeepONet to understand. To address this, we have refrained from using the term "trunk net" in the introduction and have instead provided a detailed explanation. Thanks for your valuable feedback.

---

> ### Author Response · Authors · 2023-11-23
> **Response to Reviewer XsTS - 2**
>
> > **Q4.** The paper states that the method can deal with periodic boundary conditions. How about other ones (e.g., Dirichlet and Neumann)?
>
> **A.** Thank you for pointing out an important aspect of our method. Utilizing the structure of DeepONet, as mentioned in the manuscript, enables us to enforce the boundary condition $B[u] = 0$ as hard constraints. To elaborate further, we impose hard constraints for periodic and Dirichlet boundary conditions through a modified trunk net, which is one of the significant advantages of the DeepONet model structure, as also explained in [1]. For instance, in our paper, we specifically address enforcing periodic boundary conditions in the domain $\Omega$. To achieve this, we replace the network input $x$ in the trunk net with Fourier basis functions $\left(1,\cos(\frac{2\pi}{|\Omega|}x), \sin(\frac{2\pi}{|\Omega|}x),\cos(2\frac{2\pi}{|\Omega|}x),...\right)$, naturally leading to a solution $u(t,\boldsymbol{x})$ ($\boldsymbol{x}\in \Omega$) that satisfies the $|\Omega|$-periodicity. As depicted in Figure 7, the results reveal that while other models fail to perfectly match the periodic boundary conditions, GraphDeepONet successfully aligns with the boundary conditions.
>
> While our paper primarily focuses on periodic boundary conditions, it is feasible to handle Dirichlet boundaries as well using ansatz extension as discussed in [6,9]. If we aim to enforce the solution $\widetilde{u}(t,\boldsymbol{x})=g(\boldsymbol{x})$ at $\boldsymbol{x}\in \partial \Omega$, we can construct the following solution:
> $$
> \widetilde{u}(t,\boldsymbol{x})=g(\boldsymbol{x})+l(\boldsymbol{x})\sum_{j=1}^p \nu_j[\boldsymbol{f}_{0:N-1}^1,\Delta t] \tau_j(\boldsymbol{x}),
> $$
> where $l(\boldsymbol{x})$ satisfies
> $$
>     \left\{ \begin{array}{l}
>     l(\boldsymbol{x}) = 0, \quad \boldsymbol{x} \in \partial \Omega, \\
>     l(\boldsymbol{x}) > 0, \quad \text{others}.
>     \end{array}
>     \right.
> $$
> By constructing $g(\boldsymbol{x})$ and $l(\boldsymbol{x})$ appropriately, as described, we can effectively enforce the Dirichlet boundary conditions as well. While the expressivity of the solution using neural networks may be somewhat reduced, there is a trade-off between enforcing boundaries and expressivity. We have included this content in the main text and Appendix C.6.
>
> > **Q5.** By looking at Figure 3, the proposed method gives a non-smooth solution in time and space. Why does it happen and how to improve the situation?
>
> **A.** Thank you for pointing out the aspect we overlooked. In Figure 3, GraphDeepONet was trained with a parameter setting of $K=25$ for Burgers data where it divided time $t$ into 250 intervals within the range $t\in[0,4]$. The model received inputs accordingly and predicted the solutions for the next 25 frames based on the solutions of the preceding 25 frames. While this grouping strategy led to good accuracy, the results in Figure 3 show a noticeable discontinuity every 25 frames as your concern.
>
> However, as we added results in Figure 9 in Appendix C.8, grouping frames into smaller units with $K=5$ results in a smoother prediction appearance although the error is slightly increased compared to the $K=25$ case. This trade-off in GraphDeepONet is an essential aspect of our method. We appreciate your critical observation, and we have incorporated this discussion into the paper to provide a more comprehensive explanation.
>
> > **Q6.** Minor points (Typos and using MPNN).
>
> **A.** Thank you for pointing out the details in the paper. I appreciate your feedback on the typos, and I have addressed them. Additionally, I have replaced the term "MPNN" with "MP-PDE" to refer to the model proposed in [4] to avoid confusing. Thank you.
>
> *[1] Pathak, Jaideep, et al. "Fourcastnet: A global data-driven high-resolution weather model using adaptive fourier neural operators." arXiv preprint arXiv:2202.11214 (2022).*
>
> *[2] Goswami, Somdatta, et al. "Physics-informed deep neural operator networks." Machine Learning in Modeling and Simulation: Methods and Applications. Cham: Springer International Publishing, 2023. 219-254.*
>
> *[3] Liu, Burigede, et al. "A learning-based multiscale method and its application to inelastic impact problems." Journal of the Mechanics and Physics of Solids 158 (2022): 104668.*
>
> *[4] Brandstetter, Johannes, Daniel Worrall, and Max Welling. "Message passing neural PDE solvers." arXiv preprint arXiv:2202.03376 (2022).*

---

### Official Review · Reviewer_Y4J4 · 2023-11-01

**Soundness:** 2 fair
**Presentation:** 2 fair
**Contribution:** 3 good
**Rating:** 3
**Confidence:** 5

**Summary:**

This paper proposed a graph-based DeepONet to learn a data-driven operator that can infer an approximation to the solution at arbitrary query points. The presentation feels disjointed for different parts of the paper. The theoretical contributions are arguably minor versus those in Brandstetter et al. ICLR 2022 and Lanthaler et al. TrMA 2022, and the neural architecture is almost exactly the same with VIDON (Prasthofer et al. arXiv:2205.11404, page 3 to page 4), while experimentally performs worse than Lee-Cho-Hwang ICLR 2023 in Burgers' benchmark. Currently, this paper does not read as an ICLR caliber paper.

**Strengths:**

- One of the key changes from coefficient based on initial (previous) input and "basis" being spatial-temporal to coefficient varies time to time, and the "basis" fixed. This is a mathematically well-motivated (similar to semi-discretization for functions in Bochner spaces), and meshes well with traditional reduced-order methods for spatiotemporal PDEs.
- The latent representations (lots of channels) are updated in an autoregressive fashion, instead of doing so for the predicted solution (single channel for a scalar function). Taking advantage of the natural architectural advantage (though references are missing).

**Weaknesses:**

- The key theorem, Theorem 1, is almost identical to Theorem 3.1 in the VIDON paper, even how it is proved through breaking the error down using a split taking advantages of pseudo-inverses. The only addition to Theorem 3.1 in VIDON paper is the summation in time. However, for all the PDEs with the first order time derivatives, the natural Bochner space is $L^2([0, t], V)$ where $V$ is a Sobolev space spatially, due to various energy laws. This makes how the solutions are aggregated in time in Theorem 1 is not quite right.
- The coefficients are aggregated using an attention-like product, yet in the UAT proved later, it is not proved that this choice of coefficient has the capacity to combine the "basis" get the error bound. A more handily different form is used on page 16 before definition 2. $c_j$ is out of nowhere in the proof of Lemma 2 as well, the $|\cdot |_{\ell^p}$ goes undefined as well.
- In the proof of Theorem 2, it reads "$\mathcal{G}_{\text{graph}}$ in Definition [ ]".
- The proof of Theorem 2 uses a very special grid to show the undefined $\mathcal{G}_{\text{graph}}$ above cannot approximate the solution with a very special initial condition. Given this grid, and the 50-50 chance $\mu$ associated with $f_1$, $f_2$ constructed on this grid, the new model will still fail to approximate.
- While I understand the $\{\tau_j(\cdot)\}$ are mostly referred to as "basis" in DeepONet literature, this feels quite uncomfortable. Unlike FNO having a natural basis, in DeepONet, there is no effort to actually make this set a basis. Especially, it is long known in the transfer learning community, a large number of channels in latent, especially expanded from a few initial channels, introduced a lot of redundancy.
- On page 6, it says "since the GraphDeepONet use the trunk net to learn
the global basis, it offers a significant advantage in enforcing the boundary condition $B[u] = 0$
as hard constraints". This does not make sense at all: even for the simplest homogeneous Dirichlet BC that $u=0$ on $\partial \Omega$, given any $\boldsymbol{x}\in \partial \Omega$, if the BC are "hard constraints", then $\tau_j(\boldsymbol{x})$ should be exactly 0, and there is no such MLP that can do this.
- Missing references and/or baseline comparison:
    - NN-based ROM/POD for time-dependent PDEs. Equation (3) is the same with the Karhunen-Loeve decomposition used in POD, e.g., $u(t, x) - u(t_0, x)$ is approximated by $\sum_{j=1}^p a_j(t) \tau_j(x)$ where $\tau_j(x)$ are spatial functions corresponding to the POD modes and $a_j(t)$ are the amplitudes of these modes in time.
    - Galerkin projection method for PDE operator learning using Transformers.
    - Efforts to make FNO work for arbitrary collocation points, such as Vandermonde NO (arXiv:2305.19663), NUFFT-based FNO (arXiv:2212.04689).
    - Latent state dynamics/marching schemes, e.g., Yin et al, ICLR 2023.

### Minor things
- Page 3: I have rarely seen people using $\dfrac{\partial^2 u}{\partial \boldsymbol{x}^2}$ to represent the Hessian in PDE textbooks, e.g., please check Evans Appendix A.3 and A.5.
- Page 3: if $\mathcal{G}$ represents a mapping between function spaces, then the notation "$\mathcal{G}: u^0(\boldsymbol{x}) \mapsto u^k(\boldsymbol{x})$" is technically incorrect.

**Questions:**

- Page 5: How $\phi$ and $\psi$ are constructed is quite unclear, for example, given a different set of "sensor" points, does the user have to give the adjacency matrix as well? and how the adjacency matrix are multiplied? What quantifies as the "neighboring" nodes?
- VIDON paper does not have a spatiotemporal architecture, how the authors change VIDON to ensure a fair comparison with the new proposed model?
- Throughout the paper, the function in consideration has periodical boundary condition. On page 6, it says "GraphDeepONet can enforce periodic boundaries", please explain how enforcing PBC is possible.
- The time extrapolation comparison figure has a very weird artifact for GraphDeepONet.

---

> ### Author Response · Authors · 2023-11-23
> **Response to Reviewer Y4J4 - 1**
>
> We appreciate your constructive comments. Your comments gave us a good opportunity to review and verify the contents of the paper. We summarize your comments and respond to your concerns below.
>
> > **Q1.** 1) The theoretical contributions are arguably minor versus those in Brandstetter et al. ICLR 2022 [1] and Lanthaler et al. TrMA 2022. 2) the neural architecture is almost exactly the same with VIDON [3]. 3) Experimentally performs worse than Lee-Cho-Hwang ICLR 2023 [4] in Burgers' benchmark.
>
> **A.** It seems that you may have some misunderstandings about this aspect. First, in terms of theoretical contribution, the paper [1] does not provide any theoretical analysis of graph-based models. The theoretical analysis of approximating operators with DeepONet and related discussions on error estimates are well summarized in [2]. In [3], a new model called VIDON based on DeepONet is proposed, and the error estimate for this model is proven based on the theories in [2]. Our proposed GraphDeepONet model also aims to demonstrate that the GraphDeepONet still maintains the universal approximation property using the theories related to DeepONet in [2], similar to [3]. However, our intention is not to make a significant theoretical contribution. Additionally, although the neural network architecture may appear similar to VIDON, we have focused on creating a new model using GNN with a focus on time-dependent PDE. It's important to note that our experimental setup is entirely different from the one in [4]. In [4], only the solution at 1 second after the initial condition $w(t=0,\boldsymbol{x})$ of the Burgers equation, denoted as $w(t=1,\boldsymbol{x})$, is considered. Since it is not an experiment on the evolving solution over time, it cannot be compared as having worse performance than [4].
>
> > **Q2.** How the solutions are aggregated in time in Theorem 1 is not quite right.
>
> **A.** We are grateful for the careful feedback on the solution Space. We acknowledge the reviewer's observation concerning a potential discrepancy between the solution space used by our Theorem and the space associated with the actual time-dependent PDE solutions.
>
> We would like to clarify that our research is currently centered on autoregressive models within a discrete-time framework, both from theoretical and experimental perspectives. Our primary aim is to predict solutions at specific discontinuous time points. We find your suggestion to explore an approximator capable of continuous-time predictions quite compelling. Moreover, we agree that investigating estimates within the Bochner space, as you proposed, could significantly enhance the utility and applicability of our work. We intend to carefully consider these directions in our future research.
>
> > **Q3.** The coefficients are aggregated using an attention-like product, yet in the UAT proved later, it is not proved that this choice of coefficient has the capacity to combine the "basis" get the error bound.
>
> **A.** We appreciate your feedback regarding the clarity of the relationship between our proposed decoder and Lemma 2 as presented in the paper. To address this, we have included additional text that creates a cross-reference between them. Our argument shows that the proposed decoder is designed to effectively approximate the discrete transform.
>
> In our previous manuscript, in Theorem 3.5 as referenced in [2], we discussed the configuration of a trunk network within $ \mathcal{D}_{2}$ as introduced in our study. We have now ensured consistency by also referencing this theorem within the proof of Lemma 2.
>
> > **Q4.** In the proof of Theorem 2, the new model will still fail to approximate.
>
> **A.** We greatly appreciate the reviewers' insights in identifying potential ambiguities in our examples. Upon revisiting our application of Lemma 3.6 and Theorem 3.7 from [2], we concur that the examples presented did not sufficiently present the superiority of our proposed model over conventional graph models.
>
> To rectify this, we have revised a proof of the theorem through the construction of a rotational mapping in a periodic domain. The rotation speed increases with distance from the domain's center. Conventional message-passing neural networks, which rely on consistent relative distances for information exchange, are incapable of learning this mapping due to the inherent periodicity; identical inputs at different grid points would erroneously yield the same output. Our approach deviates from this limitation by learning a global latent basis that facilitates the generation of distinct values for identical inputs across various positions. A more detailed proof and discussion are presented in Appendix B.

---

> ### Author Response · Authors · 2023-11-23
> **Response to Reviewer Y4J4 - 2**
>
> > **Q5.** Why the $\tau_j(\cdot)$ is referred to as "basis" in DeepONet, unlike FNO having a natural basis.
>
> **A.** Thank you for pointing out the lack of detailed explanation in the manuscript. It is appreciated. Considering $\tau_j(\cdot)$, the output of the trunk net in DeepONet, as a basis is an essential aspect, as acknowledged in previous studies [4,5,6,7,8]. It plays a crucial role in approximating the spatial variable $\boldsymbol{x}$ of the target function $u(t,\boldsymbol{x})$ by taking it as the input for the trunk net and effectively constructing bases for the target function space. A notable feature of DeepONet is its ability to perform operator learning by adjusting the scale of these bases depending on the input function.
>
> I am curious about your comment that DeepONet does not make an effort to create a natural basis, unlike FNO having a natural basis. As mentioned in Theorem 4.2 of [8], DeepONet learns the basis through the trunk net from the data during training, allowing the potential to acquire a more suitable representation from the data. On the other hand, FNO considers DeepONet's trunk net to be fixed, representing a trigonometric basis. It would be helpful to have more details on the reviewer's point for further advancing the paper. Thank you once again for pointing out valuable aspects.
>
> > **Q6.** Explanation on the enforcing the boundary condition $B[u]=0$
>  as hard constraints such as the simplest homogeneous Dirichlet BC that $u=0$ on $\partial\Omega$ given any $\boldsymbol{x}\in\partial\Omega$, or periodic boundary condition which is considered in experiment.
>
> **A.** Thank you for pointing out an important aspect of our method. Utilizing the structure of DeepONet, as mentioned in the manuscript, enables us to enforce the boundary condition $B[u] = 0$ as hard constraints. To elaborate further, we impose hard constraints for periodic and Dirichlet boundary conditions through a modified trunk net, which is one of the significant advantages of the DeepONet model structure, as also explained in [1]. For instance, in our paper, we specifically address enforcing periodic boundary conditions in the domain $\Omega$. To achieve this, we replace the network input $x$ in the trunk net with Fourier basis functions $\left(1,\cos(\frac{2\pi}{|\Omega|}x), \sin(\frac{2\pi}{|\Omega|}x),\cos(2\frac{2\pi}{|\Omega|}x),...\right)$, naturally leading to a solution $u(t,\boldsymbol{x})$ ($\boldsymbol{x}\in \Omega$) that satisfies the $|\Omega|$-periodicity. As depicted in Figure 7, the results reveal that while other models fail to perfectly match the periodic boundary conditions, GraphDeepONet successfully aligns with the boundary conditions.
>
> While our paper primarily focuses on periodic boundary conditions, it is feasible to handle Dirichlet boundaries as well using ansatz extension as discussed in [6,9]. If we aim to enforce the solution $\widetilde{u}(t,\boldsymbol{x})=g(\boldsymbol{x})$ at $\boldsymbol{x}\in \partial \Omega$, we can construct the following solution:
> $$
> \widetilde{u}(t,\boldsymbol{x})=g(\boldsymbol{x})+l(\boldsymbol{x})\sum_{j=1}^p \nu_j[\boldsymbol{f}_{0:N-1}^1,\Delta t] \tau_j(\boldsymbol{x}),
> $$
> where $l(\boldsymbol{x})$ satisfies
> $$
>     \left\{ \begin{array}{l}
>     l(\boldsymbol{x}) = 0, \quad \boldsymbol{x} \in \partial \Omega, \\
>     l(\boldsymbol{x}) > 0, \quad \text{others}.
>     \end{array}
>     \right.
> $$
> By constructing $g(\boldsymbol{x})$ and $l(\boldsymbol{x})$ appropriately, as described, we can effectively enforce the Dirichlet boundary conditions as well. While the expressivity of the solution using neural networks may be somewhat reduced, there is a trade-off between enforcing boundaries and expressivity. We have included this content in the main text and Appendix C.6.
>
> > **Q7.** Add baseline comparison
>
> **A.** Thank you for suggesting for additional baseline comparison. Our focus is on comparing our model with existing graph neural network(GNN)-based models capable of simulating time-dependent PDEs on irregular domains, such as MP-PDE and MAgNet. Consequently, instead of considering variations of FNO, we concentrated on GNN-based PDE solvers for experiment baseline. Therefore, FNO, being a fundamental model in operator learning area, was compared only on regular grids. We included experiments comparing our model with F-FNO proposed in [10], which is state-of-the-art on regular grids and applicable to irregular grids. As shown in Table 4, the F-FNO is applicable to irregular grids of N-S equation data, but it generally exhibits higher errors compared to GraphDeepONet. This is attributed to the limited capacity for the number of input features. We used the F-FNO model, which is built for Point Cloud data, to predict how solutions will evolve over time. At first, the model was designed to process dozens of input features, which made it difficult to include all the initial values from a two-dimensional grid. We added this explanation in Appendix C.3.

---

> ### Author Response · Authors · 2023-11-23
> **Response to Reviewer Y4J4 - 3**
>
> > **Q8.** Page 5: How $\phi$ and $\psi$ are constructed is quite unclear
>
> **A.** Thank you for accurately pointing out the detail we inadvertently omitted in the paper. As briefly mentioned in the early part of Section 3.3, the edges $(i,j)\in\mathcal{E}$ are constructed based on the proximity of node positions, connecting nodes within a specified distance. In actual experiments, we considered nodes as grids with given initial conditions. There are broadly two methods for defining edges. One approach involves setting a threshold based on the distances between grids in the domain, connecting edges if the distance between these grids is either greater or smaller than the specified threshold value. Another method involves utilizing classification techniques, such as the $k$-nearest neighbors ($k$-NN) algorithm, to determine whether to establish an edge connection. We determined whether to connect edges based on the $k$-NN algorithm with $k=6$ for 1D, $k=8$ for 2D.
>
> Therefore, based on these edges, the processing of $\phi$ and $\psi$ takes place. The crucial point here is that once the Graph $ G = (\mathcal{V}, \mathcal{E}) $ is constructed according to a predetermined criterion, even with a different set of "sensor" points, $\phi$ and $\psi$ remain unchanged as processor networks applied to the respective nodes and their connecting edges. We have added an explanation for this in Section 3.3 and the appendix.
>
> > **Q9.** How the authors change VIDON to ensure a fair comparison with the new proposed model?
>
> **A.** Thank you for pointing out a crucial aspect. As you mentioned, existing DeepONet-based models, including VIDON, generally treat the time variable $t$ equivalently to the spatial variable $\boldsymbol{x}$ without separate consideration. Thus, they utilize both $t$ and $\boldsymbol{x}$ as inputs to the trunk net, learning and predicting the solution $u(t,\boldsymbol{x})$ only within the predefined time domain $t\in[0,T]$. From this perspective, our GraphDeepONet stands out by employing GNN for the propagation of the solution concerning the time variable $t$. As evident from the experimental results, including Figure 3, when trained solely on the time domain $t\in[0,T]$, models like DON and VIDON exhibit prediction inaccuracies for subsequent time intervals. In contrast, GraphDeepONet effectively learns variations in PDE solutions over time, showcasing enhanced proficiency in predicting outcomes for time extrapolation.
>
> > **Q10.** Figure 3 has a very weird artifact for GraphDeepONet.
>
> **A.** Thank you for pointing out the aspect we overlooked. In Figure 3, GraphDeepONet was trained with a parameter setting of $K=25$ for Burgers data where it divided time $t$ into 250 intervals within the range $t\in[0,4]$. The model received inputs accordingly and predicted the solutions for the next 25 frames based on the solutions of the preceding 25 frames. While this grouping strategy led to good accuracy, the results in Figure 3 show a noticeable discontinuity every 25 frames as your concern.
>
> However, as we added results in Figure 9 in Appendix C.8, grouping frames into smaller units with $K=5$ results in a smoother prediction appearance although the error is slightly increased compared to the $K=25$ case. This trade-off in GraphDeepONet is an essential aspect of our method. We appreciate your critical observation, and we have incorporated this discussion into the paper to provide a more comprehensive explanation.

---

> ### Author Response · Authors · 2023-11-23
> **Response to Reviewer Y4J4 - 4**
>
> > **Q11.** Missing references and Minor things (Notations)
>
> **A.** Thank you for bringing to our attention relevant papers that we had not considered, along with suggesting additional references. Your recommendations have been incorporated into the paper, broadening the understanding of our work. We have added references to variations of neural operator (NO) models and papers on operator learning using transformers (Vandermonde NO, NUFFT-based FNO). Additionally, we included references to papers exploring the simulation of time-dependent scenarios using latent states (ROM, POD, and implicit neural representation). Your guidance has been greatly appreciated. Moreover, we have addressed the awkward notations you highlighted (e.g., $\frac{\partial^2u}{\partial \boldsymbol{x}^2}$ and $u^0(\boldsymbol{x})\mapsto u^k(\boldsymbol{x})$) throughout the entire manuscript. Thank you for bringing it to our attention.
>
>
>
> *[1] Brandstetter, Johannes, Daniel Worrall, and Max Welling. "Message passing neural PDE solvers." arXiv preprint arXiv:2202.03376 (2022).*
>
> *[2] Lanthaler, Samuel, Siddhartha Mishra, and George E. Karniadakis. "Error estimates for deeponets: A deep learning framework in infinite dimensions." Transactions of Mathematics and Its Applications 6.1 (2022): tnac001.*
>
> *[3] Prasthofer, Michael, Tim De Ryck, and Siddhartha Mishra. "Variable-input deep operator networks." arXiv preprint arXiv:2205.11404 (2022).*
>
> *[4] Lee, Jae Yong, C. H. O. SungWoong, and Hyung Ju Hwang. "HyperDeepONet: learning operator with complex target function space using the limited resources via hypernetwork." The Eleventh International Conference on Learning Representations. 2022.*
>
> *[5] Hadorn, Patrik Simon. "Shift-deeponet: Extending deep operator networks for discontinuous output functions." ETH Zurich, Seminar for Applied Mathematics, 2022.*
>
> *[6] Lu, Lu, et al. "A comprehensive and fair comparison of two neural operators (with practical extensions) based on fair data." Computer Methods in Applied Mechanics and Engineering 393 (2022): 114778.*
>
> *[7] Meuris, Brek, Saad Qadeer, and Panos Stinis. "Machine-learning-based spectral methods for partial differential equations." Scientific Reports 13.1 (2023): 1739.*
>
> *[8] Kovachki, Nikola, Samuel Lanthaler, and Siddhartha Mishra. "On universal approximation and error bounds for Fourier neural operators." The Journal of Machine Learning Research 22.1 (2021): 13237-13312.*
>
> *[9] Lu, Lu, et al. "Physics-informed neural networks with hard constraints for inverse design." SIAM Journal on Scientific Computing 43.6 (2021): B1105-B1132.*
>
> *[10] Li, Zongyi, et al. "Fourier neural operator with learned deformations for pdes on general geometries." arXiv preprint arXiv:2207.05209 (2022).*

---

### Official Review · Reviewer_Rs7H · 2023-11-06

**Soundness:** 3 good
**Presentation:** 3 good
**Contribution:** 2 fair
**Rating:** 5
**Confidence:** 2

**Summary:**

This paper combines GNN with DeepOnet for dynamic modeling, empirically shows better predictive accuracy, and shows some theoretical analysis on theoretical guarantee.

**Strengths:**

The authors demonstrate the ability to extrapolate which DeepONet cannot do and show this approach can achieve better predictive accuracy compared to another graph-based method and is able to better enforce boundary conditions (due to DeepONet formulation).

**Weaknesses:**

It's not clear why we want to have DeepONet + GNN. It seems that the main modification is having a DeepONet-inspired decoder to get the solution at any spatial point. It would be nice to have the intuition or insights behind this. experiments-wise - one 1D example and one 2D example also seems not as strong as other paper published.

**Questions:**

1. I am curious about the intuition and reason that this method outperforms another graph-based approach on irregular grids - such as MeshGrpahNet.
2. how are the input graph edges formed?
3. how is the boundary condition enforced like Ricker or Neumann or periodic enforced?

---

> ### Author Response · Authors · 2023-11-23
> **Response to Reviewer Rs7H - 1**
>
> We appreciate your earnest comments. Your comments gave us a good opportunity to review and verify the contents of the paper. Please see below where your comments and the corresponding responses are summarized.
>
> > **Q1.** It's not clear why we want to have DeepONet + GNN. It would be nice to have the intuition or insights behind this. The reason that this method outperforms another graph-based approach on irregular grids.
>
> **A.** The intuition and insight behind the integration of DeepONet and GNN in GraphDeepONet can be considered from two main perspectives: 1) a viewpoint of existing DeepONet and its modified models, and 2) a perspective of existing PDE solver models utilizing GNN. Initially, from the DeepONet perspective, there was a question on treating time $t$ naively as the input to the trunk net, treating it on par with the positional variable $\boldsymbol{x}$. The insight arose from the realization that the trunk net of DeepONet creates bases depending solely on $\boldsymbol{x}$, while simultaneously requiring a different approach to provide information about the changes in the solution over time $t$. Therefore, as the reviewer suggested, applying time $t$ propagation using GNN enabled the previously challenging time-interpolation.
>
> From the perspective of GNN-based PDE solvers, these models could predict solutions for irregular grids, but the results seemed to vary depending on the positioning of the irregular grid. It was recognized that predictions were limited to the predetermined grid. This was the insight and starting point of our GraphDeepONet research. To address this, as suggested by the reviewer, a DeepONet-inspired decoder was employed. More precisely, leveraging the global basis created by DeepONet's trunk net, GraphDeepONet effectively gathers information from each irregular grid, making predictions possible at all points in the domain. As you mentioned, GraphDeepONet doesn't outperform in all irregular grid cases. However, as seen in Table 2, we observed that GraphDeepONet demonstrated relatively consistent prediction accuracy.
>
> > **Q2.** Experiments-wise - one 1D example and one 2D example also seems not as strong as other paper published.
>
> **A.** Thank you for the suggestion. During the review process, we added experiments on a more complex dataset: 2D Navier-Stokes equations recommended by Reviewer 9eTr. We utilized data from the setting of the 2D Navier-Stokes equations for a viscous, incompressible fluid in vorticity form on the unit torus, as employed in [1,2]. As can be seen in Table 2, our approach exhibits similar trends for the N-S equation across different datasets. While other methods show variations in errors depending on the type of irregular grid, GraphDeepONet demonstrates good stability across irregular grids.
>
> > **Q3.** How are the input graph edges formed?
>
> **A.** Thank you for accurately pointing out the detail we inadvertently omitted in the paper. As briefly mentioned in the early part of Section 3.3, the edges $(i,j)\in\mathcal{E}$ are constructed based on the proximity of node positions, connecting nodes within a specified distance. In actual experiments, we considered nodes as grids with given initial conditions. There are broadly two methods for defining edges. One approach involves setting a threshold based on the distances between grids in the domain, connecting edges if the distance between these grids is either greater or smaller than the specified threshold value. Another method involves utilizing classification techniques, such as the $k$-nearest neighbors ($k$-NN) algorithm, to determine whether to establish an edge connection. We determined whether to connect edges based on the $k$-NN algorithm with $k=$6 for 1D, $k=8$ for 2D. We have added an explanation for this in Section 3.3 and the appendix.

---

> ### Author Response · Authors · 2023-11-23
> **Response to Reviewer Rs7H - 2**
>
> > **Q4.** How is the boundary condition enforced?
>
> **A.** Thank you for pointing out an important aspect of our method. Utilizing the structure of DeepONet, as mentioned in the manuscript, enables us to enforce the boundary condition $B[u] = 0$ as hard constraints. To elaborate further, we impose hard constraints for periodic and Dirichlet boundary conditions through a modified trunk net, which is one of the significant advantages of the DeepONet model structure, as also explained in [3]. For instance, in our paper, we specifically address enforcing periodic boundary conditions in the domain $\Omega$. To achieve this, we replace the network input $x$ in the trunk net with Fourier basis functions $\left(1,\cos(\frac{2\pi}{|\Omega|}x), \sin(\frac{2\pi}{|\Omega|}x),\cos(2\frac{2\pi}{|\Omega|}x),...\right)$, naturally leading to a solution $u(t,\boldsymbol{x})$ ($\boldsymbol{x}\in \Omega$) that satisfies the $|\Omega|$-periodicity. As depicted in Figure 7, the results reveal that while other models fail to perfectly match the periodic boundary conditions, GraphDeepONet successfully aligns with the boundary conditions.
>
> While our paper primarily focuses on periodic boundary conditions, it is feasible to handle Dirichlet boundaries as well using ansatz extension as discussed in [3,4]. If we aim to enforce the solution $\widetilde{u}(t,\boldsymbol{x})=g(\boldsymbol{x})$ at $\boldsymbol{x}\in \partial \Omega$, we can construct the following solution:
> $$
> \widetilde{u}(t,\boldsymbol{x})=g(\boldsymbol{x})+l(\boldsymbol{x})\sum_{j=1}^p \nu_j[\boldsymbol{f}_{0:N-1}^1,\Delta t] \tau_j(\boldsymbol{x}),
> $$
> where $l(\boldsymbol{x})$ satisfies
> $$
>     \left\{ \begin{array}{l}
>     l(\boldsymbol{x}) = 0, \quad \boldsymbol{x} \in \partial \Omega, \\
>     l(\boldsymbol{x}) > 0, \quad \text{others}.
>     \end{array}
>     \right.
> $$
> By constructing $g(\boldsymbol{x})$ and $l(\boldsymbol{x})$ appropriately, as described, we can effectively enforce the Dirichlet boundary conditions as well. While the expressivity of the solution using neural networks may be somewhat reduced, there is a trade-off between enforcing boundaries and expressivity. We have included this content in the main text and Appendix C.6.
>
>
>
> *[1] Kovachki, Nikola, et al. "Neural operator: Learning maps between function spaces." arXiv preprint arXiv:2108.08481 (2021).*
>
> *[2] Li, Zongyi, et al. "Fourier neural operator for parametric partial differential equations." arXiv preprint arXiv:2010.08895 (2020).*
>
> *[3] Lu, Lu, et al. "A comprehensive and fair comparison of two neural operators (with practical extensions) based on fair data." Computer Methods in Applied Mechanics and Engineering 393 (2022): 114778.*
>
> *[4] Lu, Lu, et al. "Physics-informed neural networks with hard constraints for inverse design." SIAM Journal on Scientific Computing 43.6 (2021): B1105-B1132.*

---

### Author Response · Authors · 2023-11-23
**General response to all Reviewers**

We would like to express our gratitude to all of the reviewers for their invaluable feedback and suggestions.
With the assistance of the comments, we have significantly enhanced our paper. All the suggestions raised by the reviewers were of great help in revising our manuscript, and we are pleased to note that reviewers find this study is __mathematically well-motivated__ (Reviewer Y4J4), __the theoretical analysis supports the methods' expressivity and superiority__ (Reviewer Rs7H, XsTS), and __the paper is well-written, with clear and easy-to-follow presentation__ (Reviewer 9eTr).

We are open to addressing any concerns or questions raised and welcome the opportunity to provide further clarification. It's important to highlight that modifications to the paper based on reviewers' suggestions, along with relevant links, are as follows:

* To more clearly demonstrate the strengths of our model, we __added experiments on the more complex dataset__: 2D Navier-Stokes equations [1]. Please see the response to reviewer Rs7H and 9eTr. It is contained in Table 2 of Section 4 in the main text and its corresponding explanation.
* To compare with the latest FNO model that operates on irregular domains, we have __included F-FNO [2] as a benchmark comparison__. Further details can be found in the responses from reviewers Y4J4 and 9eTr, and it is presented in Section 4 and Appendix C.3 of the main text.
* We would like to emphasize once again that our __GraphDeepONet inherits one of the advantages of DeepONet [3]__, which is __the ability to enforce boundary conditions__ that reviewers Rs7H, Y4J4, and XsTS were curious about. Although our experiments primarily focused on enforcing periodic boundary conditions, we can effectively enforce the Dirichlet boundary conditions as well. Figure 7 and detailed explanations have been added to Appendix C.6 in the main text.

Due to the limited rebuttal time, we present results for a single seed only, with general trends consistently well. For the final paper, we commit to repeating experiments over multiple seeds, mirroring the approach taken for other experiments in the main paper. For more detailed explanations, please refer to our individual responses to each reviewer. If any concerns, uncertainties, or questions from the reviewers, we are willing to address them and provide further elaboration during the discussion phase. We are delighted to engage in communication with you during the discussion period.

*[1] Kovachki, Nikola, et al. "Neural operator: Learning maps between function spaces." arXiv preprint arXiv:2108.08481 (2021).*

*[2] Tran, Alasdair, et al. "Factorized fourier neural operators." arXiv preprint arXiv:2111.13802 (2021).*

*[3] Lu, Lu, et al. "A comprehensive and fair comparison of two neural operators (with practical extensions) based on fair data." Computer Methods in Applied Mechanics and Engineering 393 (2022): 114778.*

---

### Meta-Review · Area_Chair_f3Xk · 2023-12-07

**Metareview:**

The paper explores an extension of the DeepONet architecture designed for time-dependent Partial Differential Equations (PDEs). This extension incorporates a Graph Neural Network (GNN) into the DeepONet model, replacing the branch net component. The GNN follows an encoder-process-decoder autoregressive pipeline, enabling the modeling of spatio-temporal dynamics in the latent space of the branch net. The outputs of the GNN are combined with basis functions learned from the trunk net component to compute the modeled function. This modification allows for improved time extrapolation compared to the original DeepONet. Additionally, in contrast to classical GNNs, it facilitates the computation of the modeled function at any spatial position.

The paper establishes theoretical results demonstrating the approximation capacity of the model. Experimental evaluations on 1D and 2D dynamics compare the model's performance to alternative GNN models. Reviewers acknowledge the novelty of the proposed system and recognize the advantages of the autoregressive approach. During the rebuttal, the authors addressed reviewer comments by adding a new experiment and baseline, clarifying several points, and making corrections to theoretical statements.
However, given the substantial modifications made from the initial manuscript and the limited improvements observed for certain dynamics, we believe that the paper would benefit from resubmission to another venue. This resubmission should include a revised manuscript and a strengthened experimental section.

**Justification For Why Not Higher Score:**

Several questions remain open both for the theoretical analysis and the experiments

**Justification For Why Not Lower Score:**

a

---

### Decision · Program_Chairs · 2024-01-16

Reject